# SEC-seq: association of molecular signatures with antibody secretion in thousands of single human plasma cells

Rene Yu-Hong Cheng[1,2,13], Joseph de Rutte[3,13], Cade Ellis K. Ito[1,4], Andee R. Ott[1], Lucie Bosler[3], Wei-Ying Kuo[3], Jesse Liang[3], Brian E. Hall[5], David J. Rawlings [1,6,7], Dino Di Carlo[3,8,9,10] ✉ & Richard G. James [1,2,4,7,11,12] ✉

The secreted products of cells drive many functions in vivo; however, methods to link this functional information to surface markers and transcriptomes have been lacking. By accumulating secretions close to secreting cells held within cavity-containing hydrogel nanovials, we demonstrate workflows to analyze the amount of IgG secreted from single human B cells and link this information to surface markers and transcriptomes from the same cells. Measurements using flow cytometry and imaging flow cytometry corroborate the association between IgG secretion and CD38/CD138. By using oligonucleotide-labeled antibodies we find that upregulation of pathways for protein localization to the endoplasmic reticulum and mitochondrial oxidative phosphorylation are most associated with high IgG secretion, and uncover surrogate plasma cell surface markers (e.g., CD59) defined by the ability to secrete IgG. Altogether, this method links quantity of secretion with single-cell sequencing (SEC-seq) and enables researchers to fully explore the links between genome and function, laying the foundation for discoveries in immunology, stem cell biology, and beyond.

Organisms critically depend on the proteins or other factors which cells secrete into their environment that can act locally in a paracrine manner or systemically. For example, one of the main roles of B cells is to respond to antigens with the production and secretion of large quantities of immunoglobulins targeting antigen epitopes. In this process, B cells differentiate, undergoing significant phenotypic, morphological, and genetic changes. Linking these functional changes in the capacity for secretion of immunoglobulins to genetic/ phenotypic profiles at the single-cell level can uncover the population heterogeneity and potential new cell states.

Standard single-cell analysis tools cannot simultaneously assess external cellular information (e.g., secreted proteins) with cell surface and/or intracellular information. One possible method is by combining flow cytometry with intracellular staining to quantify intracellular production of secreted proteins in the context of other markers. Intracellular proteins, including secreted proteins, can be analyzed,

[1]Center of Immunotherapy and Immunity, Seattle Children Research Institute, Seattle, WA 98101, USA. [2]Molecular Engineering and Science Institute, University of Washington, Seattle, WA 98195, USA. [3]Partillion Bioscience, Los Angeles, CA 90095, USA. [4]Department of Lab Medicine and Pathology, University of Washington, Seattle, WA 98195, USA. [5]Luminex corporation, Seattle, WA 98119, USA. [6]Department of Immunology, University of Washington, Seattle, WA 98195, USA. [7]Departments of Pediatrics, University of Washington, Seattle, WA 98195, USA. [8]Department of Bioengineering, University of California - Los Angeles, Los Angeles, CA 90095, USA. [9]Department of Mechanical and Aerospace Engineering, Los Angeles, CA 90095, USA. [10]California NanoSystems Institute (CNSI), University of California - Los Angeles, Los Angeles, CA 90095, USA. [11]Department of Pharmacology, University of Washington, Seattle, WA 98195, USA. [12]Brotman-Baty Institute for Precision Medicine, Seattle, WA 98195, USA. [13]These authors contributed equally: Rene Yu-Hong Cheng, Joseph de Rutte. ✉e-mail: dicarlo@ucla.edu; rickerj@u.washington.edu

e.g., using intracellular cytokine staining, but through a destructive process that involves permeabilization and fixation of the cell. Forming pores in the cell is necessary for fluorescently-labeled antibodies to penetrate the intracellular space and bind to chemically fixed proteins. Downsides of this permeabilization and fixation process include the loss of cell viability and loss of other intracellular molecules, such as mRNA, which limits downstream transcriptomic analysis. Furthermore, the presence of secreted proteins on or within the cell does not necessarily indicate that these proteins would have been secreted. Similarly, transcripts associated with secreted proteins may sometimes correlate to secretion[1–3]. However, transcript levels do not account for various downstream processes after transcription, including RNA splicing, translation, post-translational modifications, enzymatic cleavage, or even storage of secreted proteins in secretory vesicles prior to secretion in response to a stimulus.

Other tools to characterize cell-secreted products lack the quantitative resolution, throughput, and multiplexing of flow cytometry and do not directly link secretions to transcriptomic information. Researchers have utilized optofluidic pens or other microfluidic compartments to isolate single cells and accumulate secreted products for analysis on solid surfaces near the cells. Two recent instruments that employ these approaches are the Beacon system from Berkeley Lights, and Isoplexis' Isolight system. Both systems use microscopic imaging to analyze the secreted products from cells, with dynamic range and the number of color channels constrained by the cameras and filter sets used[4–6]. In contrast, the gold standard in single-cell analysis, flow cytometers, leverage laser-based excitation and PMT-based detection to achieve sensitive multiplexed measurements with high dynamic range. Surface markers are not readily accessible in the Isolight system, while the Beacon system can analyze a few. The number of cells that can be analyzed depends on the number of optofluidic pens or microchambers on a single chip, usually from 1,000 to 10,000. While the Beacon system can sort cells after analysis, this is not achievable with the Isolight system. Another assay format that is compatible with flow cytometry uses the cell itself to capture secreted products (e.g., the cytokine-catch assay from Miltenyi). This assay format requires specialized bispecific antibodies that have to be individually formulated for each assay (specific cytokine) and cell type (not all cells express CD45). We note that there are currently no bispecific antibody products available that bind to plasma cells and/or capture secreted IgG. Secretions can also diffuse away and bind to neighboring cells, leading to crosstalk since cells are not confined in compartments[7]. No current technology has been able to link the amount and type of secreted molecules from a single cell with the transcriptome of that same cell at a scale of hundreds to thousands of cells in parallel.

We demonstrate a technology that overcomes these tradeoffs, combining the multiplexing and high-throughput quantitative analysis of surface markers by flow cytometry or transcripts by single-cell RNA-seq with the ability to perform relative quantification of secretions (IgG) in the same single cells. The approach uses microscale hydrogel particles with a bowl-shaped cavity, called nanovials[8], which capture cells and their secretions, and are compatible with flow cytometry and single-cell sequencing instruments. We apply this technology to achieve an eight-plex multiplexed secretion assay, including six channels dedicated to cell surface markers, one channel to cell viability, and a final channel to IgG secretion. The approach is compatible with fluorescence-activated cell sorting (FACS) and imaging flow cytometry. By using oligonucleotide-barcoded antibodies, we also link IgG secretion directly to transcriptomes in the same cells by introducing nanovials containing antibody-secreting cells directly through a 10X Chromium single-cell RNA-seq workflow. We characterize the heterogeneity in the secretion of IgG across human B cell subtypes using this format and identify two subpopulations of CD38++ cells, one of which is highly secreting. Cells expected to be negative for IgG

(IgA/IgM+, activated B cells) were not found to secrete IgG using our assay. CD138 remained the best surface marker that predicted high IgG secretion. By linking secretion to transcriptomes at the single-cell level, we find several other surface markers (e.g., CD59) that have expression correlated with IgG secretion and validate the result using immunostaining. These data indicate that CD59 is expressed by IgG plasma cells that secrete large amounts of IgG. We also find that cell populations with uniformly high IgG production have increased levels of transcripts in pathways associated with mitochondria respiration, protein transport, and translation. We envision that this method could help us better understand the determinants of protein secretion in human B cells and other models.

## Results

### Workflow for measuring IgG secretion by human plasma cells

We first developed a workflow for loading and analyzing the secretions of single human B cells adhered to hydrogel nanovial particles (Fig. 1). This workflow involves capturing single cells into nanovial cavities by linking nanovials to conjugated antibodies targeted to surface proteins. After capturing, cells are incubated to facilitate the accumulation of secreted IgG onto anti-IgG antibodies precoated to the cell-associated nanovial surface. The nanovial-bound secreted IgG and other cell surface markers are then tagged with antibodies with either fluorescent labels or oligonucleotide barcodes. Labeled nanovials and cells are then analyzed using flow cytometry, imaging flow cytometry, or sorted by FACS and analyzed by single-cell sequencing.

To identify optimal surface markers to capture ex vivo-differentiated human plasmablast/plasma cells[9] into nanovials, we tested a panel of antibodies against surface proteins (CD45, CD27, and CD38) expressed in B cells and analyzed capture by flow cytometry. Nanovials are made of highly transparent hydrogel, however, the shape and larger size leads to a unique scatter signature that is readily distinguished from other cell events[10]. Cells loaded on nanovials were discriminated from free cells and empty nanovials using a combination of flow cytometry scatter and fluorescence gating. Based on the live-dead stain, only live cells were gated in the downstream analysis (Supplementary Fig. 1). We found that antibodies against CD27 yielded the highest percentage of total cells captured, and the captured cells represented a broad range of cell types, including CD19 high active B cells, as well as CD19 low IgM+ cells, IgA+ cells, and IgM/IgA double negative cells (Supplementary Fig. 1).

### Measuring IgG secretion in plasma cells expressing different cell surface markers

Previously, we found heterogeneity in antibody secretion rate for plasma cells measured using ELISPOT (Supplementary Fig. 2), however, we were unable to investigate these subpopulations further due to the inability to characterize other properties of cells in the ELISPOT format. Using nanovials, we can use flow cytometry to simultaneously assess the relative quantity of secreted IgG, along with cell surface and intracellular proteins at the single-cell level. To explore the heterogeneity in antibody secretion, we isolated B cells from peripheral blood mononuclear cells (PBMCs), and differentiated these into heterogeneous populations of plasma cells, plasmablasts, and activated B cells ex vivo[9] (see Methods for more details). After differentiation, we loaded cells onto 55 micrometer-diameter nanovials functionalized with anti-CD27 and anti-human IgG, and stained cells with a panel of B cell/plasma cell and immunoglobulin classes surface markers to define subpopulations, as well as anti-IgG for detecting secreted IgG on nanovials (Fig. 2a). A large fraction of loaded B cells exhibited a phenotype consistent with plasmablasts (PBs; CD38+) or plasma cells (PCs; CD38+CD138hi), which may represent antibody-secreting cells (ASCs). A small portion of loaded cells exhibited a phenotype of activated B cells (CD19hiCD38lo). The ASCs could further be categorized by antibody isotype. We observed subpopulations of cells with surface

expression of IgM or IgA, and double negative (DN) cells that are most likely IgG+ ASCs; few IgE cells are present in our culture system (Fig. 2b)[11].

We next associated IgG secretion with the different B cell subtypes. As expected, activated B cells and the majority of IgA and IgM cells exhibited little to no IgG secretion (Fig. 2b). To address the potential cross-reactivity and diminish the signal from free cells in the loading step, we altered the cell loading procedure to include anti-IgG antibody to block these interactions. After making these changes to the protocol, we showed that IgM+ cells had reduced IgG+ signal (Supplementary Fig. 3). A large percentage of DN cells exhibited high levels of IgG secretion (Fig. 2b, c). However, the distribution of IgG secretion in DN cells was bimodal, indicating that a large percentage of these cells did not produce IgG despite the fact that most expressed surface markers that are conventionally associated with ASCs (Fig. 2c).

To further investigate the surface markers associated with IgG secretors, we further gated DN cells based on thresholds of CD38 and CD138 (boxes, Fig. 2c), markers which increase in expression during PC maturation. Of the DN cells, we observed increased proportions of IgG-secreting cells depending on the expression of PB/PC maturation markers: low in early PBs (CD38+CD138lo; ~25% IgG), intermediate in PBs (CD38++CD138lo; ~60% IgG) and high in PCs (CD38++CD138hi; ~80% IgG, Fig. 2c). When we compared the mean fluorescence intensity of IgG secretors between CD138hi and CD38+ populations, we observed significant increases in the PCs (Fig. 2d), which indicates that while PC phenotype corresponds to a large increase in the proportion of IgG secretors, there is also a small increase in secretion amount relative to immature PBs.

We used imaging flow cytometry (Amnis ImageStream) to confirm that the anti-IgG signals we observed in PCs originated from single cells and were secreted into the associated nanovial. We analyzed loaded B cells on ImageStream (Gating strategy, Supplementary Fig. 4) and used the images to measure fluorescence on the nanovials and cells separately. The signal for secreted IgG was distinct from fluorescence in the

cell and evident as a crescent/ring shape on the inner surface of nanovials (Fig. 2e, arrowhead). Upon quantification of IgG in the other cell phenotypes, we found that CD38++CD138hi PCs exhibited significantly higher IgG+ signal relative to CD38++CD138lo or CD38+ cells (Fig. 2f and representative images Supplementary Fig. 5). Collectively, these data indicate that CD138hi PCs are the predominant source of secreted IgG in human plasma cell cultures.

## Compatibility of nanovial analysis with single-cell sequencing

We first evaluated the general compatibility of nanovials with single-cell transcriptomic sequencing using the 10X Genomics Chromium system. The Chromium system uses a microfluidic droplet generator to encapsulate cells with single barcoded hydrogel beads inside drops, where lysis and single-cell reverse transcription is performed. The gel beads contain oligonucleotides that hybridize to mRNA and other feature barcodes from the cell sample that also comprise a unique barcode that can be associated with each single cell. In our modified workflow, we simply replaced the cell sample in the microfluidic chip with nanovials loaded with cells (Fig. 3a). Nanovials with a diameter of 35 micrometers could be introduced into the microfluidic chips and entered into microfluidically generated droplets (Supplementary Fig. 6a, b). Nanovials of larger sizes could flow through the chips, but only after deforming significantly to pass through the channels (Supplementary Fig. 6c), which led to clogging in some experiments. To reduce the chance of clogging, we used 35 micrometer-diameter nanovials for the remaining single-cell sequencing experiments.

We found that mRNA from mouse hybridoma cells and human Raji cells seeded on nanovials was successfully transcribed and sequenced. The number of transcripts and transcribed genes recovered from Raji cells on nanovials was comparable to freely suspended cells (Fig. 3b). Notably, nanovials can be tagged with feature barcodes (a nanovial barcode) by coating with oligonucleotide-conjugated streptavidin. We used this feature to differentiate Raji cells loaded on nanovials from cells freely floating in suspension (Fig. 3b). Similarly,

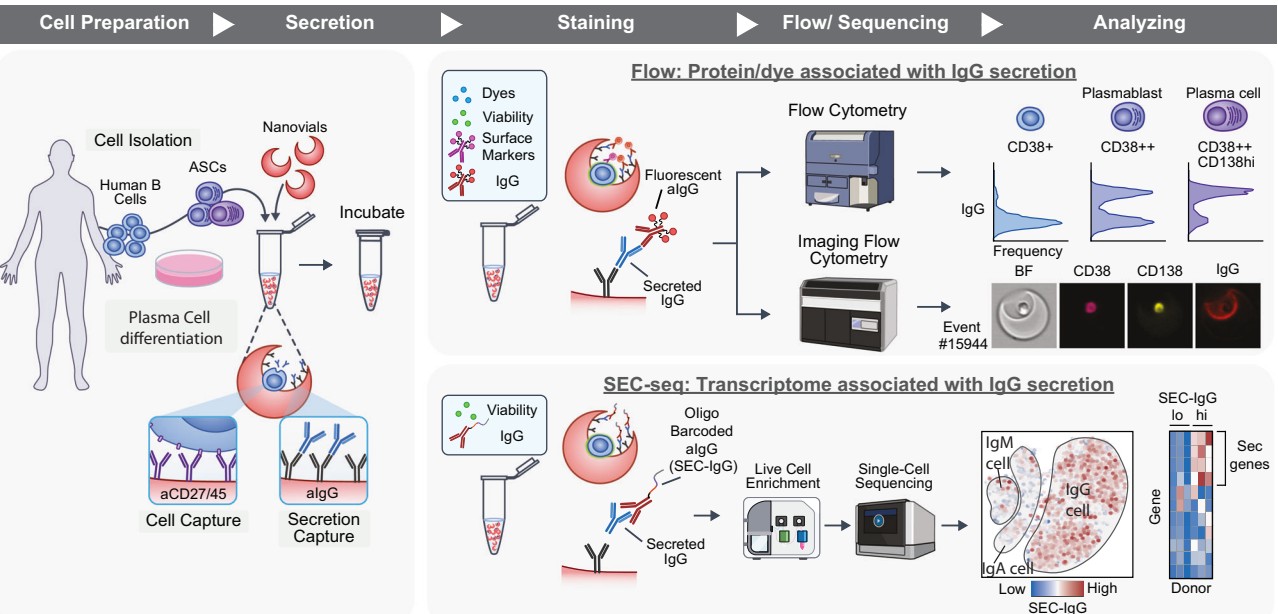

**Fig. 1 | Workflow to link IgG secretion to surface markers and transcriptomes at the single-cell level.** Human B cells are isolated from donors and expanded in a differentiation cocktail to promote differentiation into antibody-secreting cells. Cells are then loaded into a slurry of nanovials in a tube where they bind to antibodies on the nanovials for cell surface markers (CD27 or CD45). The loaded nanovials are incubated to accumulate secreted IgG on the surface via anti-IgG capture antibodies. Nanovials are then stained with fluorescent or oligo-barcoded anti-IgG, as well as viability dyes and other surface marker stains. Stained nanovials and associated cells are analyzed by flow cytometry (LSR II flow cytometer), imaging flow cytometry (ImageStream), or sorted (Nanocellect WOLF) for single-cell transcriptomics using the 10X Chromium system. Data linking IgG secretion with surface markers/functional dyes and transcriptomes at the single-cell level is acquired and analyzed.

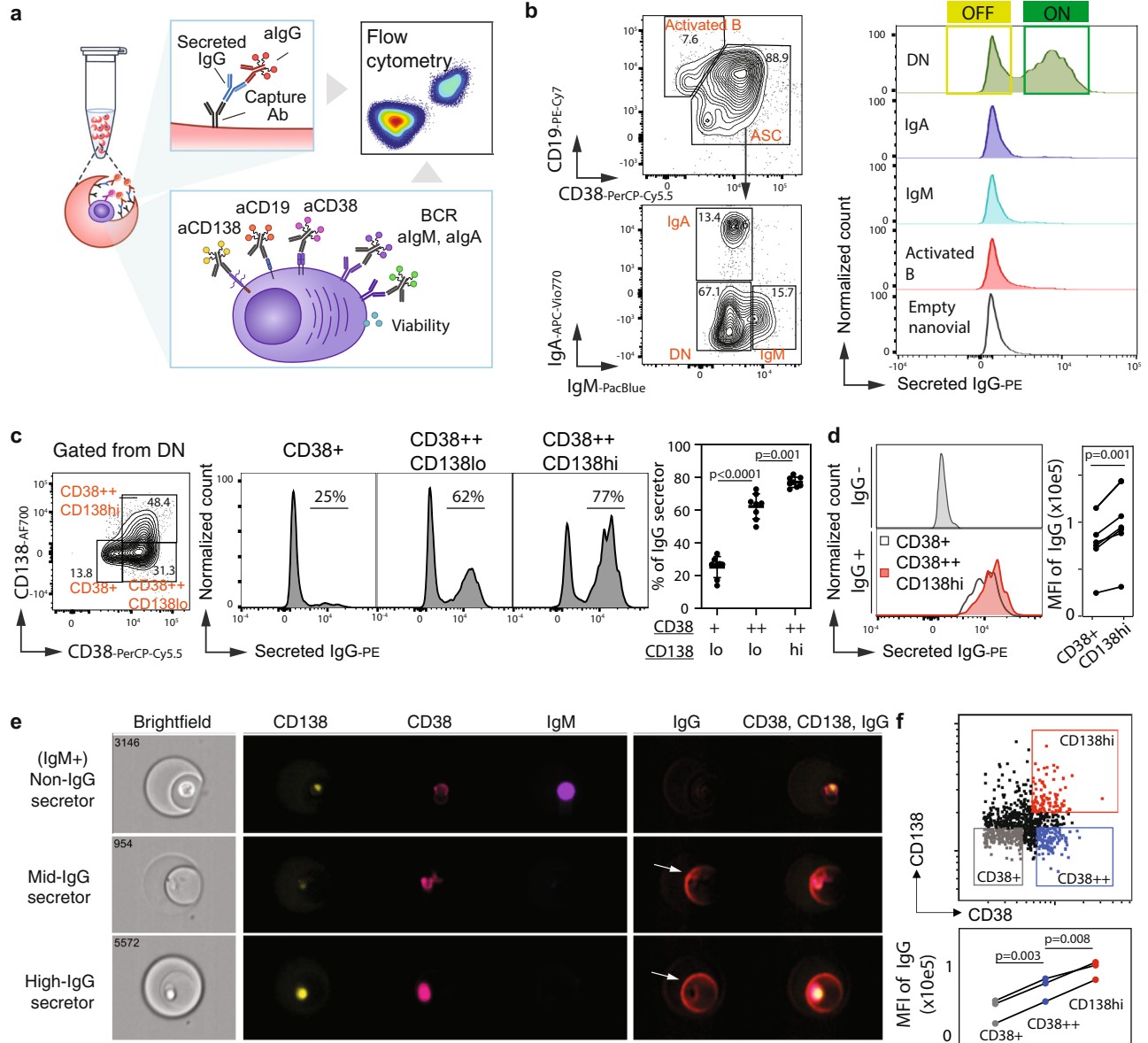

**Fig. 2 | Linking IgG secretion to cell surface markers and intracellular machinery using flow cytometry. a** Schematic of the staining format used to analyze single B cell IgG secretion and cell surface markers by flow cytometry. Created with BioRender.com. **b** Representative flow cytometry density scatter plots for surface markers to identify populations of ASCs from active B cells using CD19 and CD38 staining. IgA cells, IgM cells, and ASCs not producing either IgA or IgM (double negative, DN) were gated based on IgA and IgM staining. Fluorescence histograms of IgG secretion signal for the various identified gates and empty nanovials containing no cells. **c** Contour plot for CD38 and CD138 staining in the DN population and fluorescence histograms of IgG signal from populations within these gates. The dot graph presents the % of IgG secretors from each subset ($n = 8$ independent analyses, from four donors). Data were presented as mean ± SD. $P$ values were calculated using a paired one-way analysis of variance with a Tukey's

test for multiple comparisons. Source data are provided as a Source Data file. **d** Histogram of IgG levels for IgG non-secretors and IgG secretors from different ASC subsets. The linked dot graph presents the mean fluorescence intensity of IgG secretors from each ASC subset. The $p$ value was calculated using a two-sided paired $t$-test ($n = 7$ independent analyses, from three donors). Source data are provided as a Source Data file. **e** Imaging cytometry results confirm that IgM+ cells have low levels of secreted IgG, CD138loCD38++ populations have intermediate levels of secreted IgG, and CD138hiCD38++ have the highest levels of secreted IgG. **f** Imaging flow cytometry gating strategy for IgG quantification by cell subtype (upper). The dot graph presents the MFI of IgG from each ASC subset ($n = 3$ independent analyses from three donors). $P$ values were calculated using paired one-way analysis of variance with Tukey's test for multiple comparisons. Source data are provided as a Source Data file.

we used reads of anti-IgG associated feature barcodes to categorize hybridoma cells as off nanovial, on nanovial with low anti-IgG barcode reads, or on nanovial with high anti-IgG barcode reads (Fig. 3c). Similar to our results with Raji cells, we found a comparable number of transcripts and gene numbers recovered from hybridoma cells on nanovials when compared to free hybridoma cells, which were largely independent of the amount of IgG secretion.

We next evaluated mixing by loading mouse hybridoma cells and human Raji cells on separate batches of nanovials and mixing the

batches together before loading them into the microfluidic device. We prepared nanovial samples at the recommended concentration to recover 1000–2000 cells (10,000 nanovials assuming 10–20% cell loading). At this concentration, we found the majority of cells had single species-specific gene reads (Fig. 3d), while a small minority (1.5%) had both human and mouse reads, suggesting coincident loading of two cell types in a droplet (Supplementary Fig. 7). We searched the cDNA library for heavy and light chain sequences for the anti-hen egg lysozyme (HEL) antibody produced by the hybridomas. Sequences

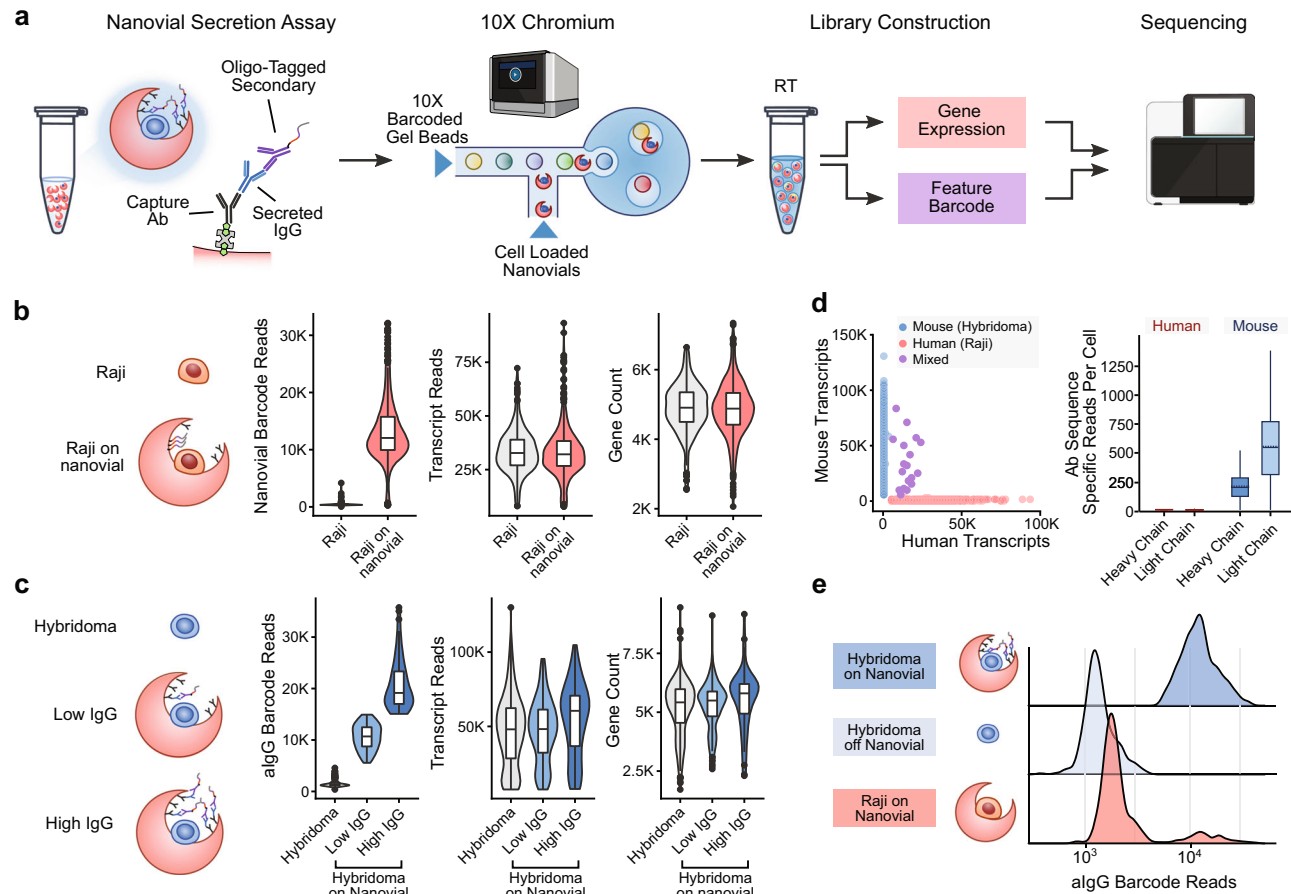

**Fig. 3 | Compatibility of nanovials with single-cell transcriptomic sequencing using the 10X Genomics Chromium system. a** Workflow for barcoding and analyzing secretions on nanovials. Captured secreted IgG is labeled with oligonucleotide-modified anti-IgG antibodies. Nanovials and associated cells are introduced into the 10X Chromium Next GEM Chip and emulsions are formed containing nanovials and Barcoded Gel Beads. Reverse transcription (RT) is performed to create cDNA and then amplified to form separate feature barcode and gene expression libraries. These libraries are sequenced and feature barcode reads are linked to each cell's transcriptomes. **b** Human Raji cells on nanovials (transcripts linked with a streptavidin feature barcode, Nanovial Barcode) are compared with Raji cells off nanovials based on the number of transcript reads and gene count.

**c** Mouse hybridoma cells that secrete IgG that are off nanovials are compared to cells on nanovials based on the number of anti-IgG feature barcode reads, transcript reads, and gene count. **d** Scatter plot of mouse vs. human transcript counts when equal amounts of human Raji and mouse hybridoma cells on nanovials are input into the Chromium system. Specific heavy and light chain antibody sequences are recovered from mouse hybridoma cells. **e** Secreted IgG feature barcode read histograms for hybridomas on nanovials, hybridomas off nanovials, and Raji cells on nanovials. For the boxplots shown in **b**–**d** center is the median value, the bounds of the box represent the interquartile range (IQR) and the whiskers denote the 1.5 X interquartile range. Dots represent outliers beyond the whiskers (*n* = 1237 cells).

were recovered from the hybridoma population and were not detected in the Raji cells, as expected (Fig. 3d). Consequently, using nanovials did not lead to increased cell-cell mixing (i.e., shared barcodes for more than one cell) compared to statistical expectations.

By adding an oligonucleotide-barcoded anti-IgG label, we could link the secretion of IgG to the mouse transcriptome for individual cells on nanovials, an approach we refer to as secretion-encoded cell sequencing (SEC-seq). As expected, the lowest number of anti-IgG feature barcode reads were associated with free cells (not loaded in nanovials) (Fig. 3e). In contrast, events with the highest IgG feature barcode reads were associated with mouse hybridomas on nanovials. As expected, most Raji cells on nanovials had low feature barcode reads, with a small group with higher reads proportional to the fraction of nanovial multiplet events (Fig. 3e and Supplementary Fig. 7).

### Simultaneous measurement of protein secretion and single-cell transcriptome sequencing (SEC-seq)

We have previously shown that ASCs will mature ex vivo, and secrete more antibodies per cell with longer culture periods[11]. To better

understand the transcriptional programs that associate with IgG secretion during ASC maturation, we exploited the SEC-seq technique to explore the transcriptomes of ex vivo-differentiated ASCs as a function of the IgG secretion phenotypes following 10 and 13 days of culture. In this workflow, we adapted the SEC-seq protocol by pre-sorting nanovials containing viable human differentiated B cells from three donors immediately prior to loading into emulsions with the 10X Barcoded Beads (Fig. 4a, and gating in Supplementary Fig. 8a, Cell Ranger QC summary in Supplementary Fig. 8b). The data from the sequencing was analyzed to simultaneously assess the degree of IgG secretion (SEC) via signal from barcoded IgG antibodies (left panel, Fig. 4b) and gene expression sequencing (right panel, Fig. 4b). Clustering of the transcriptional data was largely driven by expression of the specific antibody isotypes (Supplementary Fig. 9a). The majority of IgG-secreting cells were in clusters expressing *IGHG1*, *IGHG2, IGHG3* and *IGHG4* (Supplementary Fig. 9a), with a small minority in clusters expressing *IGHM, IGHA1*. As expected, we found cells from each donor and each time point in every cluster (Fig. 4c). However, the cells taken from the day 10 time point were enriched at the top of the UMAP, and included a cluster that was poorly

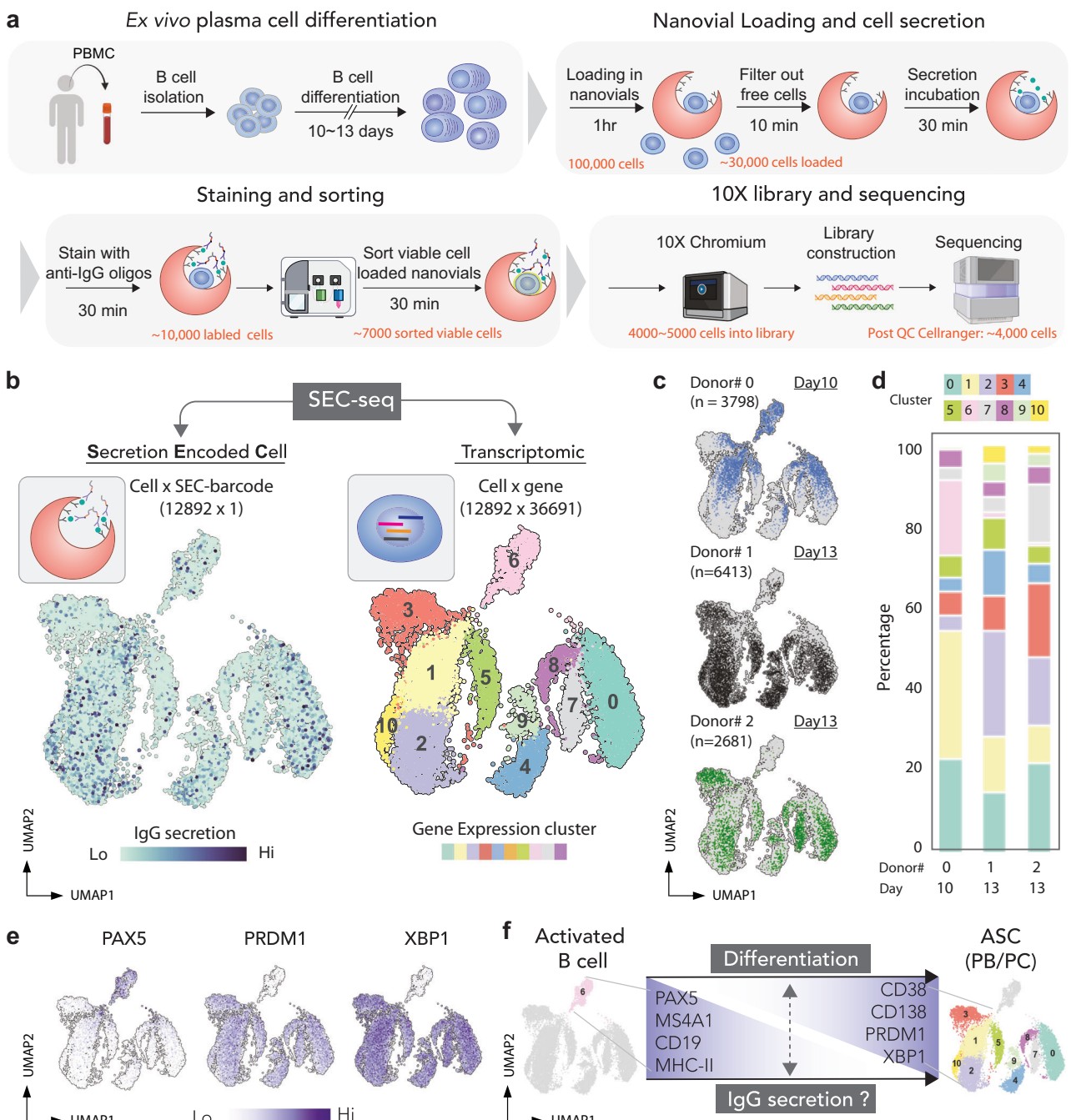

**Fig. 4 | SEC-seq: single-cell transcriptomic sequencing associated with IgG secretion. a** Workflow for SEC-seq to link IgG secretion to transcriptome at the single-cell level. Partially created with BioRender.com. **b** Transcriptome-based clustering of single-cell expression profiles of human ASCs (right), and the levels of the corresponding levels of the anti-IgG barcode projected on the same UMAP plot (left; this data is compiled from 12,892 cells derived from three independent experiments and three donors). **c** Distribution of cells from each donor overlaid onto the UMAP plot. **d** The percentage breakdown of each donor's cells represented in each cluster in the UMAP is shown. **e** Transcription factor gene expression levels for a marker of activated B cells (*PAX5*) and PCs (*PRDM1, XBP1*) overlaid on the UMAP plot and **f** illustration of canonical marker expression along the PC differentiation axis. All UMAP color bars are according to normalize expression data into units of log transcripts.

represented in the two samples prepared at day 13 (cluster 6, Fig. 4d). Upon overlaying the expression levels of genes associated with activated B cells (*CD19, MS4A1, HLA-DRA, HLA-DMA*, and *PAX5*) and ASCs (*XBP1, IRF4, PRDM1, CD38*, and *CD138*), we saw that the day 10 cells were enriched for expression of activated B cell markers, and that the day 13 cells were enriched for expression of ASC markers (Fig. 4e, f and Supplementary Fig. 9b). These data confirm that the SEC-seq procedure can effectively capture, and analyze mRNA in cells from several stages of ASC maturation.

## Using SEC-seq to determine transcriptional signatures associated with IgG secretion

To classify PCs by isotypes, we used a similar strategy as we previously used to gate double negative cells by flow cytometry. The distributions for the gene counts of IgA and IgM were both bimodal (Fig. 5a). We drew "gates" for *IGHM⁺*, *IGHA⁺* at the local minimum between the modes in each distribution and further analyzed the DN cells (Fig. 5a), which we believe are likely to express IgG mRNAs. We defined "IgG cells" as DN cells that exhibited non-zero *IGHGx* mRNA expression.

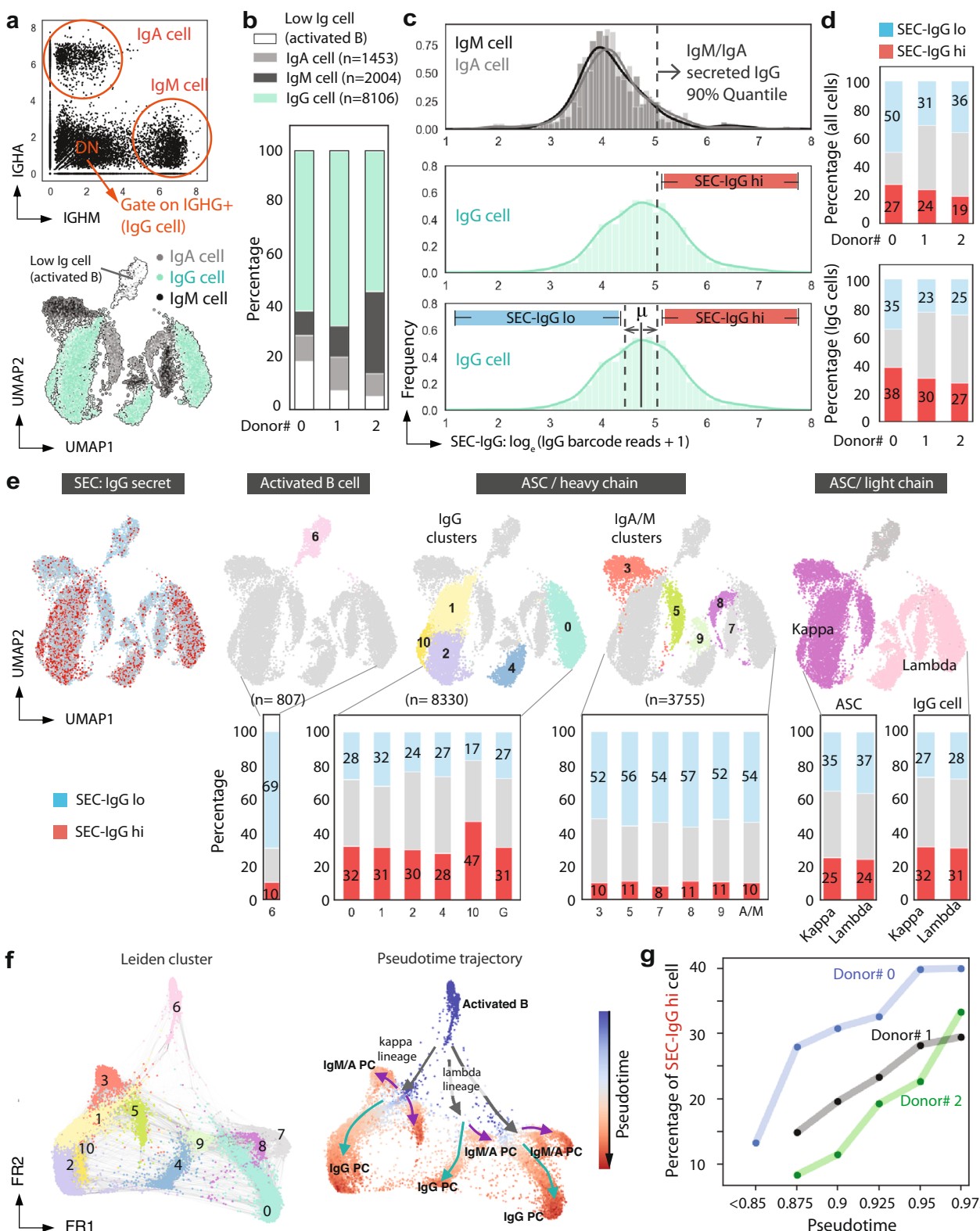

Using these gates for IgA, IgM, and IgG cells resulted in consistent clusters localized to the bottom of the UMAP with the day 10 donor having the fewest ASCs (Fig. 5a, b). In agreement with the flow cytometry data, we observed a log increase in the median number of unique IgG barcode reads in the IgG population relative to the IgM/IgA populations (Fig. 5c). Based on the distribution of IgG barcodes in the *IGHM/A*+ cells, we calculated confidence intervals and defined SEC-IgG high cells as those with a cutoff above the 90% quantile of the *IGHM/A*

barcode distribution (dashed line Fig. 5c). Then, to further segment the population, we defined SEC-IgG low cells as those with IgG barcodes fewer than the mean minus the difference between the SEC-IgG high cutoff and the mean (see bottom image, Fig. 5c). Upon applying these definitions to each dataset, we observed a similar proportion of SEC-IgG high cells across all three donors (Fig. 5d).

Next, we asked how the SEC-IgG high cells were distributed in each subcluster. After the projection of these cells onto the UMAP

**Fig. 5 | IgG secretors are associated with different gene expression clusters and pseudotime. a** IgM, IgA, and IgG expressing populations are gated based on the mRNA expression level of *IGHA* and *IGHM*. Cells with *IGHGx* transcripts are highlighted in green. Each population was projected on the gene expression UMAP plot. **b** Percentage of the total cell population from each donor following in IgM, IgA, IgG, and activated B cell categories. **c** Histograms of unique IgG barcode reads in IgA/IgM (top) and IgG (middle, bottom) subsets. Dashed lines on the bottom histogram define the thresholds for SEC-IgG hi and SEC-IgG lo cells. **d** Percentage of all cells and IgG expressing cells that are SEC-IgG hi and SEC-IgG lo. **e** SEC-IgG hi cells (red) and SEC-IgG lo cells (blue) projected onto the UMAP; Activated B cells, IgG clusters, IgA/M clusters, and light chain clusters are highlighted on the UMAP. Bar graphs

display the percentage cells in the three thresholded regions in (**c**) for each indicated cluster corresponding to the upper panel. **f** Single-cell graph by PAGA trajectory inference, and heatmap by inferred pseudotime of all cells. Trajectory starts from the initial state node (cluster 6, activated B cells from day 10 cells). Cell differentiation states in between are labeled according to their representative cell status. Heatmap vmin (minimum value to anchor the colormap) = 0.80. **g** The percentage of SEC-IgG hi cells in each 0.5 pseudotime frame (e.g., x-axis = 0.875 stands for 0.85–0.9 interval) for each donor is plotted as a function of pseudotime. The data is from 12,892 cells derived from three independent experiments and from three individual donors.

(Fig. 5e), we found that the majority of SEC-IgG high cells (red dots) overlap with IgG cell clusters, and a small minority overlap with the IgA/M clusters (see also Supplementary Fig. 9a). Upon looking at the percentage of SEC-IgG high cells in each subcluster, we found that IgG clusters have on average ~31% IgG SEC-IgG high and ~27% SEC-IgG low cells, whereas the remaining clusters were predominantly SEC-IgG low cells. The *IGHG3* (cluster 10) subclusters exhibited a higher percentage of SEC-IgG high cells, possibly indicating different levels of IgG secretion among PCs expressing these isotypes. In contrast, we found there was a similar percentage of SEC-IgG high and SEC-IgG low cells when cells expressed either light chain kappa or lambda (Fig. 5e).

We then asked if PC differentiation is associated with increased IgG secretion. To extract PC differentiation trajectories from the single cell data, we used Partition-based graph abstraction (PAGA), along with the Leiden-graph-based algorithm to cluster cells. We chose the root cluster (cluster 6; pseudotime = 0) to be that with the highest expression of the activated B cell markers *PAX5, MS4A1, CD19*, and *HLA-DRA, HLA-DMA* (Fig. 5f and Supplementary Fig. 9b); in this model differentiated PC clusters were at the branch tips (pseudotime = 1.0). Because of the large transcriptional differences between PB/PCs and activated cells[11,12], the breadth of the pseudotime is found in activated B cells and rare cells that are transitioning between the activated B and PB/PC clusters (pseudotime = 0 to 0.8), with most of the cells within the PB/PC clusters above pseudotime 0.85 (Supplementary Fig. 10). As expected, the activated B cell population had only ~10% SEC-IgG high cells (Fig. 5e–g). In contrast, in the PB/PC clusters, we found that SEC-IgG high cell percentage increased from ~10% to up to 40% (Fig. 5g) between pseudotime 0.85 and 1.0. These data indicate that along the trajectory of PC differentiation, there is an increase in the proportion of SEC-IgG high cells, which aligns with our flow cytometry data (CD38+ to CD138hi, Fig. 2).

### Using SEC-seq to find surrogate PC markers based on a definition of "High antibody secretion"

We hypothesized that we could use our SEC-seq dataset to uncover surrogate surface markers associated with the highest secreting PCs (Fig. 6a). Previously, several markers were found as PC markers by biopsying locations where PCs are thought to reside, e.g. cells from tonsil, spleen, bone marrow or multiple myeloma[13–17]. However, there have not been tools that directly link these markers with the most important function of PCs: antibody secretion.

We used SEC-seq data to analyze the correlation between the transcriptome and IgG secretion (SEC-IgG reads) among all cells. Consensus genes that show a correlation among three donors were evaluated as potential surrogate PC marker candidates (Fig. 6b). As expected, genes that have the highest positive correlation with SEC-IgG are the IgG genes (*IGHG1-4*), and those with the most negative correlation are non-IgG isotype genes (*IGHM/A*). mRNAs from known PC markers (*CD38, MZB1*, and *XBP1*) show a correlation with SEC-IgG in all three donors, suggesting that the analysis robustly identifies markers of PCs (Fig. 6b, c). Among PC transcription factors, only *SUB1* and *XBP1* show a positive correlation ($r \geq 0.1$) and not *IRF4* and *PRDM1*. A number of mRNAs for cell surface proteins also showed correlation

with SEC-IgG (*CYBA, ITM2C, CD59, TMEM59, MIF*, etc, UMAP in Supplementary Fig. 11), despite not previously being reported as PC markers[18].

CD59 is a membrane attack complex inhibitor[19,20], but little is known about its function in PC biology[21]. We evaluated CD59 expression in PCs using flow cytometry, along with other canonical PC markers CD98 (no correlation with SEC-IgG)[22,23], CD38, and CD138. After gating cells by the expression of each surface marker (low, medium, high), CD59, CD38, and CD138, but not CD98 showed an increased percentage of IgG secretors (Supplementary Fig. 12a). We further compared secreted IgG in nanovials with cells that were either CD38hiCD59hi or CD38hiCD138hi, the standard human PC phenotype and found that both marker sets exhibited similarly high proportions (~70%) of IgG secretors (Supplementary Fig. 12b).

### IgG secretion is highly regulated by mitochondrial metabolism and protein transport

To focus the analysis on genes that regulate or facilitate protein secretion, as opposed to those that regulate plasma cell antibody isotype subsets, we compared SEC-IgG high and low cells that solely express IgG mRNAs. As expected, PCs expressed only one antibody isotype and the majority of IgG cells expressed IgG1 (76.5%), followed by IgG3 (12.7%), IgG2 (8.2%), and IgG4 (2.7%) (Supplementary Fig. 13a). As implied by the cluster analysis we found that *IGHG3*-expressing PCs had a higher percentage of SEC-IgG high cells, whereas those expressing *IGHG2* had fewer (Fig. 7a). Regardless of the isotype, we found a moderate correlation between steady-state IgG transcripts and SEC-IgG barcodes (Fig. 7b).

We next identified differentially expressed genes between SEC-IgG high IgG cells and SEC-IgG low IgG cells (Fig. 7c; cell compartments for differentially expressed genes, Fig. 7f). Our analysis revealed enrichment for mRNAs for genes associated with the unfolded protein response, e.g. *SSR3, SSR4*, and *SEC61B*, known to be highly expressed in PCs[24,25], in the SEC-IgG high cells. Additionally, SEC-IgG high PCs exhibit increases in the mRNA levels for specific genes that regulate glycosylation (Fig. 7f). After applying gene enrichment analysis (GSEA) to Hallmark and biological process (Fig. 7d), we found that the most highly enriched genes were mitochondrial associated gene sets, including oxidative phosphorylation, and ATP synthesis coupled electron transport (Fig. 7d and Supplementary Fig. 13b, c). The other major enriched genes are the translation process and trafficking proteins, e.g., protein secretion, cotranslational protein targeting to membrane, and endoplasmic reticulum (ER) (Fig. 7d and Supplementary Fig. 13b, d, e). Moreover, our analysis showed that MYC-target genes are highly enriched in the Hallmark dataset (Fig. 7d and Supplementary Fig. 13b, f). Previous studies demonstrated that MYC-target genes are highly associated with ribosome function, oxidative phosphorylation, protein export, etc.[26,27]

To validate our transcriptional findings, we used functional dyes to measure mitochondrial volume, ER volume, and glucose uptake such as mitotracker, ER-tracker, and 2-NBDG. We found that PCs with high mitochondrial volumes or with high-ER content were predominantly IgG secretors (Fig. 7e, top panel and middle panel).

However, contrary to our expectations, we observed no association between 2-NBDG import and IgG secretion (Fig. 7e bottom), which suggests that glucose uptake, at least that assayed by 2-NBDG, is not a limiting step for high antibody secretion by PCs. Overall, SEC-seq has enabled us to integrate single-cell secretion and gene expression data (Fig. 7f). The results of this analysis imply that the rate-limiting determinants of antibody production/secretion are the cellular programs required for protein secretion, like protein translation, protein transport, the unfolded protein response, and cellular metabolism, rather than transcript availability.

## Discussion

By leveraging lab-on-a-particle technology, we can now simultaneously interrogate the degree of IgG secretion with surface marker expression or sequencing data at the single-cell level. In single experiments, we were able to analyze 3000–6000 ex vivo-differentiated human PCs, directly linking the degree of IgG secretion with transcriptomes in more than 10,000 total cells. This unprecedented scale of experiments enabled the characterization of secretion phenotypes in subpopulations of PCs. These data show that some conventional PC markers (e.g., CD38) and surrogate markers (CD59) are highly correlated with IgG secretion. We also found several pathways to be associated with high IgG secretion including oxidative phosphorylation, MYC targets, and the interferon response. While the correlation data also show that the IgG mRNAs (*IGHG1-4*) are among the most highly correlated with IgG secretion, especially *IGHG3*, several isotype mRNAs exhibited relatively low correlation scores (Fig. 7b, $r = 0.06-0.31$). These data reinforce the need to assay cell secretory function directly, since gene expression of the underlying protein is not highly correlated to protein levels[28] or protein secretion. Finally, our data imply that nanovial-based systems

could be a powerful tool to enrich PCs with high IgG secretion or antibody specificity.

By using standard equipment and workflows, the SEC-seq approach should be widely adoptable by other researchers, which promises to amplify the impact of the technique. Cells loaded on nanovials can be directly input into the microfluidic droplet generator used in the 10X Chromium system, and all other processes follow standard workflows for single-cell transcriptomics. Nanovials do not interfere with lysis, reverse transcription, or downstream cDNA library preparation or sequencing steps. The number of transcripts and gene count for cells on and off of nanovials remained similar (Fig. 3b) and doublet events remained comparable to those observed in cells alone (Fig. 3d). In fact, all of the components used to perform SEC-seq are commercially available including the biotinylated nanovials (Partillion Bioscience) and oligonucleotide-barcoded antibodies (BioLegend). Linking secreted protein function with specific nucleic acid sequences of heavy and light chains of IgG (Fig. 3d) or alpha and beta chains of T cell receptors (TCRs) can enable functional discovery workflows for monoclonal antibodies or engineered TCR-based therapies[6,29]. A parallel study is also exploring the transcriptome that underlies the secretion of high levels of vascular endothelial growth factor by mesenchymal stromal cells[30], which can elucidate features associated with therapeutic subpopulations of cells. We demonstrate here that the technique is compatible with standard CITE-seq[31] reagents to create workflows that link surface proteins, secreted proteins, and single-cell transcriptomes[32]. More broadly, nanovials can link surface markers and secreted proteins in viable cells selected using multicolor FACS. Here we demonstrated an eight-color panel, but like surface-marker-only panels, the scale of markers should only be limited by the spectral overlap of fluorophores.

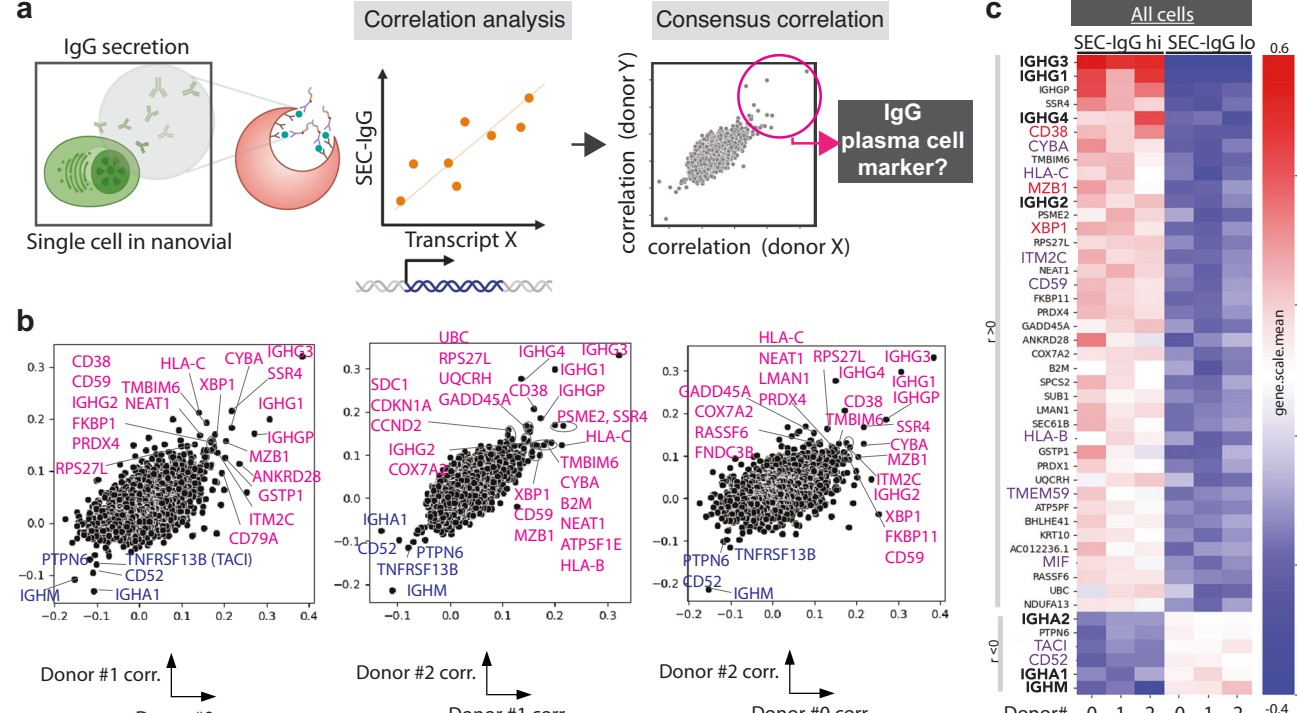

**Fig. 6 | Correlation analysis of SEC-seq data to find surrogate PC markers by IgG secretion ability. a** Schematic showing how we establish correlation scores between mRNA transcript levels and the SEC-IgG barcode. Created with BioRender.com. **b** Scatter plots showing donor-by-donor correlation scores between SEC-IgG and individual mRNA transcripts. The highest correlated genes across all donors are labeled. **c** Top ~40 highly consensus correlated genes sorted by mean of

correlation coefficient from three donors. Heatmap represents the subgroup (SEC-IgG lo, SEC-IgG hi from all cells) mean of the z-score of each gene expression from each donor. Immunoglobulin are labeled in bold, known PC markers are labeled in red, and genes expressed on cell surface are labeled in purple. The data is from 12,892 cells derived from three independent experiments and from three individual donors. Source data are provided as a Source Data file.

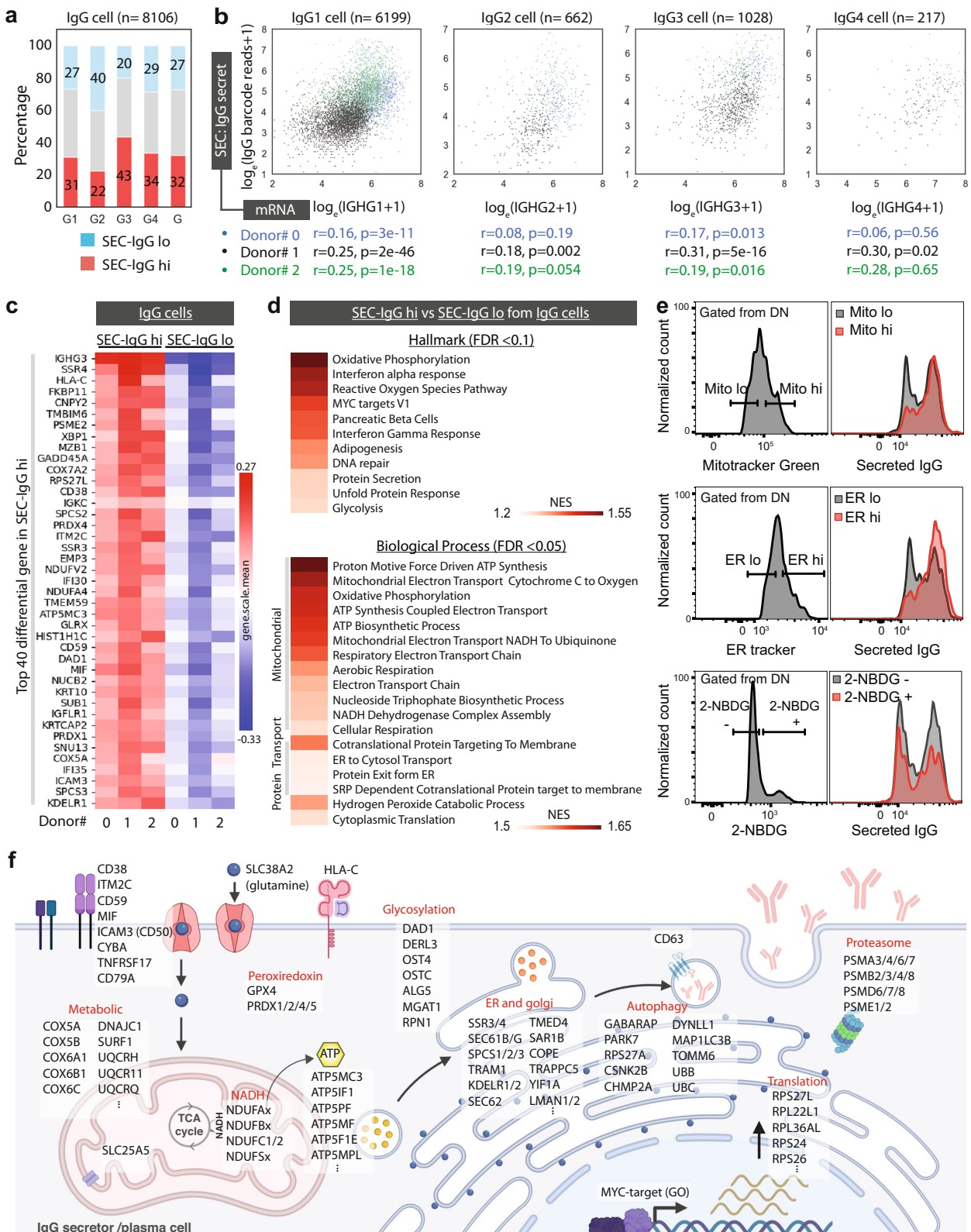

However, the SEC-seq assay used in the study still has some limitations, such as IgG isotype evaluation and a limited dynamic range. Our study found that it can be challenging to interpret the IgG isotype secretion, but it is possible to use customized barcode conjugated isotype-specific antibodies for an extended study. Because of the high levels of IgG secretion by PCs, which is estimated to be secreted at ~10,000 molecules per second[33,34], nanovial capture can become

saturated. Note that in our protocol, we capture the secreted IgG in a 30 min period, and capturing for longer periods resulted in diminished dynamic range. In similar studies quantifying interferon-gamma secretion by EBV-reactive T cell clones[35], we captured secretion for 3 h, and observed a similar or possibly a better dynamic range (~2 orders of magnitude), likely due to the lower secretory rate of interferon gamma[36]. Our study used the smallest size (35 micrometer-

**Fig. 7 | Analysis of SEC-seq data from IgG cells to identify enriched genes associated with IgG secretion. a** Bar graphs display the percentage of cells in the three thresholded regions using thresholds of SEC-IgG lo and SEC-IgG hi from each sub-isotype of IgG cells. **b** Scatter plots of *IGHG* log transcripts and log IgG barcode reads (SEC-IgG). *r* is the Pearson correlation coefficient, and the *p* value was the result of a two-tailed *t*-test relative to the null hypothesis (no correlation). **c** Top differentially expressed genes of SEC-IgG hi. Heatmap represents the subgroup (SEC-IgG lo, SEC-IgG hi from IgG cells) mean of *z*-score of each gene expression from each donor. Source data are provided as a Source Data file. **d** Gene set

enriched list in Hallmark and biological process from gene set enrichment analysis (GSEA), heatmap represents the normalized enrichment score (NES), FDR stands for false discovery rate. **e** Histograms of IgG secretion levels compared with high/low mitochondria activity, high/low ER amount, and 2-NBDG (glucose analog) uptake ±. **f** Schematic cartoon of genes upregulated in IgG secretors (adjust *p* values <0.1) based on different gene function clustering (DAVID) and GSEA. The data is from *n* = 8106 IgG cells derived from three independent experiments and from three individual donors. Created with BioRender.com.

diameter) nanovials to ensure compatibility with the 10X Chromium controller microfluidic channel (Supplementary Fig. 6b). The dynamic range can be further improved by using nanovials with an increased cavity size to increase the secretion loading capacity. Furthermore, we are working on engineering nanovials for additional applications; examples are altering stiffness to accommodate different cell types or adding reversible conjugation on nanovials to facilitate easier cell loading and/or release of the cells to clone high-performing populations.

CD38 and CD138 are well-established markers of differentiated human PCs, and long-lived PCs respectively[37–39], however, we introduce CD59 as a surrogate marker for functional PCs. CD138, (also known as SDC1) functionally regulates PC survival in the bone marrow[39,40]. However, there is no reported association between CD38 or CD138 surface expression with the degree of antibody secretion. The marker combined with CD38++CD138hi enriches for the largest fraction of IgG secretors (>75%) from DN cells. Similar to what has been observed previously[41,42], we also found that a subset of CD38low cells secrete IgG and do so at a slightly decreased degree than other IgG secretors. From the SEC-seq, we found that mRNAs for CD38 and the surface marker CD59 showed strong correlations with SEC-IgG among transmembrane proteins. We found that CD59 is more highly expressed in PCs than CD138, and in combination with CD38hi could identify ~70% of IgG secretors from all live B cells. We propose that a combination of CD59, CD138, and/or CD38 should be evaluated in future studies as a high-confidence marker of antibody-secreting PCs.

Antibody-secreting PCs exhibit unique biochemical features that enable prolific protein secretion (reviewed in refs. 43,44). From the SEC-seq analysis, *XBP1* is one of the top differential expression genes in IgG SEC-IgG high population (Fig. 7c), which suggests that increased levels of the unfolded protein response pathway may facilitate increased immunoglobulin secretion[45]. While PC-specific knockout of XBP1 eliminates immunoglobulin secretion[46] without eliciting cell death. To be activated, XBP1 transcripts are regulated at the level of splicing by the upstream unfolded protein response gene IRE1[47]. Recently, pharmacological compounds have been developed that increase[48] or decrease[49] IRE1 activity, and thus regulate the degree of spliced, active XBP1. We propose that drugs regulating the unfolded protein response factors could be leveraged to transiently increase or decrease the amount of antibodies secreted by PCs.

The pathways most highly correlated with IgG secretion were oxidative phosphorylation (OxPhos) and ATP synthesis (Fig. 7d). ATP plays the fundamental role of protein exocytosis, solubility, stability, and folding processes[50–53]. And it is essential for post-translational protein translocation through ER, and certain ATP analogs have been reported to inhibit cotranslational protein translocation, but not protein synthesis[54]. Moreover, Blimp1 (encoded by *PRDM1*) was found to be required for OxPhos in mouse plasmablasts[55]. Additionally, during differentiation from B cells to plasma cells, the levels of OxPhos increased in plasmablasts following differentiation ex vivo. Intriguingly, similar to IRE1 inhibitors, OxPhos inhibitors (oligomycin, FCCP, antimycin A) caused decreased antibody secretion without compromising cell survival[55]. In future studies, we envision using nanovials and/or SEC-seq to identify nontoxic pharmacological and/or genetic strategies that can increase and/or decrease immunoglobulin

secretion. We envision that such molecules could be used to transiently increase the antibody responses to pathogens and/or decrease antibody responses in antibody-driven disease like lupus or arthritis.

We show that nanovial technology and SEC-seq can be used to link the levels of antibody secretion to cell surface markers, transcriptional signatures, and vital dyes. This technology enables us to study the molecular determinants of protein secretion by PCs. Because high and low-secreting cells can be captured by cell sorting, we predict that this technique will be used to study PC secretion at a single cell level and co-evaluate a myriad of parameters including, but not limited to epigenetics, metabolomics, signaling, loss of function and mutational scans. Beyond the analysis of secreted immunoglobulins, the technique can unlock the ability to study >3000 proteins that are part of the human secretome[56], ultimately identifying shared, or unique, molecular underpinnings of secretion pathways that are critical for cell communication and function from the single-cell to the organismal level.

## Methods

### Ethical statement

We adhere to all applicable ethical regulations in our research. Deidentified human PBMCs were acquired under informed consent from the Fred Hutch Specimen Processing and Research Cell Bank (protocol #3942). Total of six donors were involved in flow/imaging cytometry experiments; four donors are females with an age range of 22 to 38 years old and two donors are males ages 23 and 27 years old. All donors from SEC-seq are females with an age range of 22 to 38 years old. Sex was not considered in the study design or analysis.

### Cell culture methods

All cells were cultured in incubators at 37 °C and 5% $CO_2$ in static conditions unless otherwise noted.

**Human primary B cells.** We isolated B cells from healthy donors' peripheral blood mononuclear cells (Fred Hutchinson Cancer Research Center) using the EasySep Human B cell isolation kit (Stem Cell Technologies). Isolated B cells were cultured in Iscove's modified Dulbecco's medium (Gibco) supplemented with 2-mercaptoethanol (55 μM) and 10% FBS. Cells were cultured for 7 days (activation) in medium containing 100 ng/mL megaCD40L (Enzo Life Science), 1 μg/ml CpG ODN2006 (Invitrogen), 40 ng/ml IL-21 (Peprotech), and then for 3 to 6 days (plasmablast/plasma cell differentiation) in medium containing 50 ng/ml IL-6 (Peprotech),10 ng/ml IL-15, and 15 ng/ml interferon- α2B (Sigma-Aldrich).

**Hybridoma cells.** HyHel-5 cells were maintained in IMDM media (Invitrogen) supplemented with 10% FBS (Invitrogen #16000044) and 1% penicillin/streptomycin (Invitrogen). Cells were passaged down to a final concentration of $4 \times 10^5$ cells/mL every three days.

**Raji cells.** Cells were sourced from ATCC (CCL-86™) and maintained in RPMI-1640 ATCC modification media (Invitrogen A1049101) supplemented with 10% FBS (Invitrogen) and 1% penicillin/streptomycin (Invitrogen). Cells were passaged down to a final concentration of $4 \times 10^5$ cells/mL every 3 days.

## Methods of coating nanovials

Nanovials were coated with streptavidin by mixing equal volumes of streptavidin (300 µg/mL) and nanovials (n = 340,000) for 15 min at room temperature followed by three washes which consisted of resuspension in clean washing buffer (supplementary Table 1) and centrifugation at 400×g. The coating antibody mix was prepared by making a 10X dilution of biotinylated anti-IgG (0.5 mg/mL) and biotinylated anti-cell surface protein (anti-CD27 or anti-CD45) (0.5 mg/mL) in washing buffer (final antibody concentration 0.05 mg/mL per antibody). Equal volumes of the coating antibody mix and the streptavidin-coated nanovials were combined and placed at room temperature for 30 min or 4 °C overnight. These antibody-coated nanovials were washed twice in a washing buffer. After the final wash, the supernatant was removed and the nanovials were resuspended in cell media. The final prepared nanovials were placed on ice until cell loading.

## Methods of cell loading in nanovials

Differentiated human B cells were first processed by ficoll density gradient centrifugation to remove dead cells and debris. For secretion experiments, 5 µl of blocking antibody (anti-IgG, Southern Biotech) was added to the 50 µl cell solution to yield a final concentration of 25 µg/mL. Then 55 µl of the cell solution (244,000 cells) was mixed with 20 µl of concentrated nanovials (n = 340,000) at a ratio of 1:1.4 (cell: nanovial) on ice by pipetting for 30 s with a circular motion. Care was taken to pipette cells throughout the pellet of nanovials to enable optimal loading. We then added 1 ml of biotin-free medium and incubated the mixed cells and nanovials on ice for 1 h without any perturbation.

## Methods of incubation of cells with nanovials

After the 1 h loading process, samples were placed on top of the strainer and washed with 1 ml wash buffer two times. Prepared a 15 ml falcon tube precoated with washing buffer, and flipped the strainer on top of the tube. About 2 ml of washing buffer was used to wash the nanovials off the strainer and into the falcon tube. We then centrifuged the 15 ml falcon tube containing the filtered nanovials at 300×g at 4 °C for 5 min, and resuspended nanovials into a medium pre-warmed to 37 °C. The falcon tubes containing nanovials in media were then incubated at 37 °C on top of a rotator operating at a speed of 10 rpm for 30 min to accumulate secretions on nanovials.

## Methods of cell staining of surface markers and secreted IgG

FACS tubes were precoated with staining buffer (supplementary Table 1). After incubation to capture secretions, nanovial samples in falcon tubes were then centrifuged at 400×g for 5 min and resuspended into FACS tubes with a cocktail of antibodies (supplementary Table 2, with 200-fold dilution for all antibodies, except CD138-AF700 with 100-fold) to stain cell surface markers and secreted IgG in staining buffer. Samples were stained on ice for 20 min, then washed twice, before analysis by flow cytometry.

## Methods of flow cytometric analysis

Flow cytometric analysis was performed on an LSR II flow cytometer (BD Biosciences) and events were analyzed using FlowJo software (v10.8.1). Flow cytometry gating for fluorescent proteins and viability, and immunophenotyping can be found in supplementary Figures.

## Methods of imaging flow cytometric analysis

An ISX 493 MKII equipped with 405, 488, 560, 592, 642, and 785 lasers were used to interrogate B cell secretion capture on nanovials. 40000 objects were collected under 20x magnification with laser power and channel assignments as listed in Supplementary Tables 3, 4.

Quantification of IgG signal on nanovials from cell secretions was performed through the following gating strategy illustrated in Supplementary Fig. 4. First, objects in focus were identified by the GradientRMS function in the brightfield channel. Single nanovial objects were separated from debris and multicellular aggregates based on an Aspect Ratio vs Area. Visual inspection of objects located in the gates drawn in S4B confirms the correct placement of gate settings. CD38/CD138 double-positive cells were further identified via the intensity feature on the CD38 and CD138 channels and separated into high and low signal populations. The amount of IgG secretion for these four populations was quantified by constructing a "nanovial mask" by subtracting the area localized to cells (CD38 or CD138) from the mask identified in the brightfield channel.

## Methods for visualizing nanovials through 10X Genomics Chip G

Fluorinated oil with surfactant, gel bead solution and nanovial solutions were added into the reservoirs of the 10X Genomics Chip G. 3 mL syringes (Becton Dickinson) were connected to the bead and sample inlet reservoirs via PEEK tubing (IDEX) and a coupler molded out of PDMS. Syringe pumps (PhD 2000, Harvard Apparatus) were used to inject air into the reservoirs and pressurize the bead and sample inlets and drive flow. Droplet formation videos were recorded using an inverted microscope (Nikon TE300) equipped with a high-speed camera (ZWO ASI144MM).

## Mixed species SEC-seq validation experiment

Barcoded nanovial preparation (Raji sample). Nanovials (4 million/mL) were modified by mixing a solution of nanovials in washing buffer with a cocktail of 40 µg/mL streptavidin and 20 µg/mL TotalSeqC conjugated streptavidin (Biolegend, 405271) solution at equal volumes. The sample was then placed on a rotator at room temperature for 30 min. Nanovials were washed three times by centrifuging the sample at 200×g for 5 min, removing the supernatant, and resuspending in the washing buffer. Following washing, the streptavidin-modified nanovial solution was then mixed with a solution containing 8 µg/mL anti-human CD45 (Biolegend, 304004) at equal volume. The nanovials were then placed on a rotator at 4 °C overnight. The nanovials were then washed three times as above and resuspended in cell media before proceeding with cell experiments.

Non-barcoded nanovial preparation (Hybridoma sample). Nanovials (4 million/mL) were modified by mixing with a solution of 60 µg/mL streptavidin at equal volume following standard procedures as described above. The streptavidin-modified nanovial solution was then mixed with a solution containing an antibody cocktail comprising 8 µg/mL anti-mouse CD45 (R&D Systems, AF114) biotinylated using EZ-Link Micro Sulfo-NHS-LC-Biotinylation Kit (Thermo 21935) and 20 µg/mL Goat anti-Mouse IgG FC (Jackson Immuno Research, 115-065-071) at equal volume.

Raji and Hybridoma cells were loaded separately into nanovials on ice and incubated for 1 h to allow binding. Background cells were removed using a 20-µm cell strainer and cells were then incubated in a CO2 incubator at 37 °C for 30 min to accumulate secreted IgG. The hybridoma sample was stained with a 12 µg/mL solution of TotalSeqC conjugated anti IgG1 antibody (Biolegend, 406636) and incubated for 45 min on Ice. Samples were then washed three times with ultra-pure PBS containing 0.04% BSA (Invitrogen, AM2616). The Raji and Hybridoma samples were mixed at a 1:1 ratio prior to proceeding with single-cell sequencing library preparation. Libraries were prepared at the UCLA sequencing core using the 10X Genomics Chromium Next GEM Single Cell 5′ Kit v2 + Feature barcode libraries. Approximately 8000 nanovials were added in the sample lane and emulsified with barcoded beads with the Chromium instrument. Libraries were QC'd using the tapestation (Agilent) and sequenced using NovaSeq S2 (100 Cycles). Fastq files were processed by cellranger and the Barnyard reference database (refdata-gex-GRCh38-and-mm10-2020-A) was used for analysis. A custom reference library was constructed to search for the known Ab sequence of the hybridoma cell line. Final analysis and plotting was performed using Loupe Browser (v6.0) and Seurat.

## SEC-seq human ASCs sample preparation and sequencing

Following ex vivo differentiation, cells were treated with Ficoll to remove any debris. Viable cells were immediately loaded onto nanovial and incubated to accumulate secretions, as described above. After accumulating secretions on the nanovials, samples are stained with PE anti-IgG (BD), and a secondary antibody (anti-PE) that was labeled with barcoded oligonucleotides (BioLegend). Next, we sorted viable cells loaded on nanovials with the WOLF cell sorter (nanocellect) by gating of negative SYTOX green signal (Supplementary Fig. 8). Sorted cell-loaded nanovials were introduced into the 10X Genomics Chip G (10X Genomics), at 2500-10000 cell-loaded nanovials per lane. Next, we prepared libraries using 10X Genomics Chromium Next GEM Single Cell v3.1 kit following the 10X user guide (CG000317). Libraries from the oligonucleotide-barcoded antibodies bound to nanovials and transcripts were evaluated by tapestation (Agilent) before sequencing. Finally, libraries were pooled at a ratio of 80% cellular RNA to 20% oligonucleotide-barcoded antibodies and sequenced with NextSeq 1000/2000 kit (Illumina) using the following read length: 28 bp Read1, 10 bp i7 Index, 10 bp i5 Index, and 90 bp Read2.

## Single-cell RNA-seq analysis

Fastq files were processed by cellranger based on the human reference genome GRCh38. The h5 file was then further analyzed using a custom python script. Data analysis including normalized, batch correction, dimensional reduction, hierarchical clustering, Leiden clustering, and differential gene analysis are analyzed by scanpy and BBKNN[57,58]. Figures were plotted by matplotlib (3.5.1) or seaborn (0.12.2).

## ELISpot (enzyme-linked immunospot)

PBS pre-wetted 96 well plates (Millipore) were coated overnight at 4 °C with goat anti-human IgG (H + L) capture antibody (Jackson Immunoresearch). About 100 l IMDM was used to block each well for 2 h at 37 °C. A designated number of cells were washed with PBS and resuspended in a 2X cytokine cocktail in IMDM. We added 100 l of cells to the elispot plate directly and incubated at 37 °C in an incubator overnight. Cells were then washed away and plates were washed six times. Secreted IgG was detected by binding to HRP-conjugated goat anti-human IgG secondary antibody (Southern Biotech). Spots were developed with AEC Substrate Kit and Peroxidase (HRP) reagents (Vector Laboratories). The spots were quantified using the Cellular Technology Limited; CTL ImmunoSpot software.

## Statistics and reproducibility

Sample sizes in each experiment are described in each figure and legend. Flow cytometry and SEC-seq experiments are designed to have at least three biological replicates from independent donors. Most clustering and gene differential analysis were using scanpy (1.8.2). Correlation analysis is using SciPy (1.8.0). Dataframe-based analysis and table output is generated by pandas (1.4.2). Arrays and basic statistic (mean, sum) is calculated by numpy/pandas (1.21.5). Histogram and kernel-density plot are generated by matplotlib (3.5.1). Statistical analysis was performed by scipy.stats, Graphpad or excel. Dot graphs were presented as mean ± SD by Graphpad Prism 8 (GraphPad, San Diego, CA). In SEC-seq, low-quality cells or cells with more than one light chain/heavy chain isotype were excluded from prior analysis. There was no a priori information available to make the groups non-random. The Investigators were not blinded to allocation during experiments and outcome assessment.

## Reporting summary

Further information on research design is available in the Nature Portfolio Reporting Summary linked to this article.

## Data availability

The SEC-seq single-cell read data in this study are available under the BioProject "PRJNA953084") and from GEO under accession number "GSE229042". Source data are provided with this paper.

## Code availability

The data were analyzed using published software packages and scripts. A python notebook was used to call routine statistical functions and to organize the data. In addition, the python source code used to perform data analysis is available from GitHub at (https://github.com/Rene2718/SEC-seq_plasma-cell_nanovial)[59].

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

## Acknowledgements

We thank 10X Genomics for providing reagents for these studies. We thank Nanocellect for providing access to a WOLF Sorter to perform sorting, and Donna Munoz for her assistance in these efforts. We would like to thank Brotman Baty Institute and the UCLA Technology Center for Genomics & Bioinformatics for performing sequencing services. We thank Jason Debley, Lucille Rich, Timothy Cherry, and Lulu Callies for

offering and helping with the 10X chromium controller and bioanalyzer. Last, we would like to thank Huiyun Sun in Fred Hutch for offering help for library QC. Funding: This work was supported in part by the Seattle Children's Research Institute (SCRI) Program for Cell and Gene Therapy (PCGT), the Children's Guild Association Endowed Chair in Pediatric Immunology (to DJR), the Hansen Investigator in Pediatric Innovation Endowment (to D.J.R.), the NIH under numbers 5R01CA201135 (to R.G.J.), R01AI140626 (to R.G.J.), and NIH grant 1R43GM144000 (to J.d.R.).

## Author contributions

R.Y.-H.C., J.d.R., D.D.C., and R.G.J. designed the study and wrote the manuscript. D.J.R. and R.G.J. advised the B cell experiments and design. A.R.O. and R.Y.-H.C. cultured human B cells. R.Y.-H.C. performed the IgG secretion experiment with flow cytometry and analyzed the data. B.E.H. ran samples on the ImageStream and assisted in the downstream analysis of the collected data. J.d.R., D.D.C., and W.-Y.K. designed the initial SEC-seq validation experiments which were performed by L.B. and W.-Y.K., and analyzed by J.L and J.d.R. R.Y.-H.C., J.d.R., D.D.C., and R.G.J designed the SEC-seq human plasma cell experiment. C.E.K.I. helped to sort live cell-loaded nanovials. R.Y.-H.C. performed the SEC-seq experiment. R.Y.-H.C. made the 10X library and analyzed the SEC-seq data.

## Competing interests

D.D.C., J.d.R., L.B., W.-Y.K., J.L., and the University of California have financial interests in Partillion Bioscience. J.d.R., L.B., W.-Y.K., and J.L. are employees of Partillion Bioscience. The remaining authors declare no competing interests.
