## [Peer Review File · Nature Communications]

REVIEWER COMMENTS

Reviewer #1 (Remarks to the Author):

Summary: In this work, the authors develop a method to link secreted material from a cell (here, antibodies) with the standard 10X single cell sequencing readout. Overall the work is well done and clearly presented, and I think this technique could be useful for many applications. A few minor clarifications could be helpful.

Major Points:

--I don't feel like we have a mechanistic explanation for how NADH level alterations (or any aspect of mitochondrial biology) change antibody secretion. Can the authors speculate on this a bit? Overall (e.g. Figure 6) the authors link IgG secretion levels to several different pathways (glucose uptake, mitochondria, etc.) Some validation that the genes in these pathways functionally influence IgG secretion would be helpful.

--I'm curious to know whether the authors could have obtained the same information by simply measuring cytoplasmic/ER or surface-bound IgG protein rather than encapsulating the secreted antibody in a hydrogel. A comparison to the enriched gene sets observed by cytoplasmic/ER expression (antibody binding) vs. secreted IgG would be interesting in any case to determine whether there are really genes that selectively influence secretion mechanism (rather than just translation)

Minor Points:

--Fig. 1: a bit more detail in labeling would help here. for example, what are M, A, G populations in the lower right?

--In the single cell seq data, did the authors observe correlation between high IgG transcript level and high IgG protein level? Is there a strong correlation when the authors look at their data? Is there any previous literature on the correlation between IgG transcript level and protein level? this would help us understand the level of noise in their workflow.

Reviewer #2 (Remarks to the Author):

Cheng and colleagues present an article entitled 'SEC-seq: Association of molecular signatures with antibody secretion in thousands of single human plasma cells'. Herein, the authors describe a technique that combines secretion with transcriptomic analysis. This is very interesting as a potential technique for B cell and antibody research, and the interest goes beyond – the simplicity of the workflow is promising. Therefore, in terms of technology, the approach is novel and promising and might be of interest to the readership of Nature Communication. The authors study ex vivo stimulated B cells from healthy donors and correlate a proxy of secretion with the transcriptome of the cells. These experiments recapitulate findings from the B cell field but also show some interesting new findings. Here, however, the experiments performed and presented seem preliminary and, while showing interesting initial findings, are far from displaying conclusive biological data and the vigor of data I would expect from a publication in Nature Communications.

Firstly, many experiments seem to be done with one donor (Figure 1d) or two donors (Figure 1c) only. This statement is also extracted from the conclusion (as the authors state that 3000 cells can be analyzed in a run, and Figure 4c shows 3060 cells) that the experiment has been performed once on one donor. This drastically limits the biological meaning of the results.

Second, the introduction states, 'No current technology has been able to link the amount and type of secreted molecules from a single cell with its transcriptional profile, nor has this type of association been multiplexed at a scale of hundreds to thousands of single cells simultaneously.' This statement is not exactly valid as it neglects a variety of work from different research groups and companies – the authors present the system from Berkeley Lights and Isoplexis, but many publications have been published over the last years linking the level of secretion to the sequences (which is a similar level of quantitation as the study provides). Therefore, I wonder if the presented technology really overcomes these trade-offs. While still innovative and interesting, this, in my view, decreases the added value of the method. A combination of flow cytometry and nanovials has already been published.

Thirdly, the authors should clarify in their manuscript whether they study 'human plasma cells', primary or ex vivo stimulated. As it stands, it is often left open to interpretation, and what we can learn from each system is fundamentally different.

Additionally, no comments about efficiencies are performed – if I start with 1000 B cells secreting antibodies, what can I expect at the end? If I do the math with the information provided, only a small fraction will be analyzed successfully. This would be a major limitation (or at least a point) worth mentioning and discussing. Over the figures, it is very difficult to understand how efficient the whole process is, from a cell culture of a certain number of cells to the transcriptomic end. In general, many figures describe the process (or show examples/illustrations) but give little quantitative information – this is necessary if you want to convince readers to use this technique. Many relevant controls are missing – what is the frequency of ASCs in your ex vivo stimulated cells? Does this agree with your measurements? How were these findings controlled?

I also challenge the statement that this technique quantitates secretion – I am pretty convinced that you can see differences between secreting and non-secreting, but the text often refers to these two subclasses only and also says from time to time 'level of secretion'. A level and quantitation are very different, and many more controls are needed to make such a statement (as various techniques allow quantitative secretion analysis down to the molecules/s). Throughout the manuscript, the authors call cells low and high secreting – it seems not clear why you have two groups of cells, and looking at figure 5c, it looks more like a continuous log Normal distribution and not two distinct populations. Also, the legend calls the cells (non-secreting), whereas the figure and text refer to the same cells as low IgG – if they are low IgG, this would also mean that most IgA and IgM are low IgG secreting? Here, I would rather suggest that some IgG-secreting cells and cells do not secrete. Also, there is also a considerable amount of IgM and IgA cells above the threshold of log5 – what does this mean?

In summary, while I appreciate the simple approach and the value of the technique, considerably more experiments, and controls are needed to conclude the biological data. As the experiments will take some time, I would reject the current version of the manuscript, but I remain available if the data is added.

A few minor comments from my side:

- I think it needs to be clear that these are ex vivo stimulated human B cells – this makes a huge difference as we would expect a single Gaussian distribution in terms of secretion (and one
- Why are viabilities so low? In our experience, you should expect 80-90% viability after 10 days of ex vivo stimulation, yet the data shown is somewhere between 14-50 % for loaded vials – is this viability sufficient for sequencing? I understand that only viable cells are sorted, but such an environment will certainly have an influence on secretome and transcriptome.
- Figure 1c: Why one-way instead of two-way? The figure also shows 6 points, the legend 5.
- 'Upon looking at the percentage of IgG secretors in each subcluster, we found that IgG clusters have on average ~ 25% IgG secretors and ~ 40% non-secretors, whereas the remaining clusters were predominantly non-secretors.' How does this correlate with the statement from before that between 20-70% of cells were ASCs in the cytometry experiments?
- 'Approximately 12% of the loaded nanovials were IgG positive in the cell sort (Fig. S8a, right panel), which is comparable to the ~14% of IgG secretors we quantified in the scRNA sequencing data.' – 12% are positive in the sort, but 25 % are IgG secretors in the subclusters? How does this work?
- As the Raji/Hybridoma experiments show multiple loading (cross-contamination) of up to 9%, what does this mean for the sequencing experiments of B cells? Should I expect a similar background of multiplets?
- Acknowledges: 'other members' – either names or remove

- If I calculate the numbers in SI Figure 1b, I get much lower frequencies – please double check. Also, the viability is very low – do the nanovials fish out mostly dead cells or is the general viability very low in the culture? Both have implications for your results... this is not discussed.
- As a remark to Figure S2: Larger spots could also be to larger cells? I know that this is often done in ELISPOT, but I am always wondering about the potential bias in these analyses – why are two classes displayed? How are they defined? Should an integral of a larger spot not anyway be larger if not corrected by the size?
- Figure S3: without axes and nothing, what do I see?
- Nature Portfolio Report: How can you calculate sample sizes of 1 and 2, or 3'000? How was this done?
- Gating strategy: How was the gate for living/dead chosen? Why does CD19 expression in the gate vary so highly?
- Lastly, the flow cytometry, in contrary to the reporting chapter, are not displaying axis, numbers and fluorophore (for example, figure 1b, 1c, 1d).
- Within the discussion, data are discussed that are not shown – please remove this statement or show the data. Lastly, the discussion also does not discuss the potential limitations of the technique.

Reviewer #3 (Remarks to the Author):

In their previous studies, James and colleagues developed bowl-shaped hydrogel particles, termed “nanovials”, that allow capturing of secreted proteins from single cells and are compatible with flow cytometry analyses. In the current manuscript, the authors extended previous work to show that the nanovial system is compatible with the commercially available single-cell mRNA sequencing workflow. They further provided proof-of-principle demonstration of the SEC-seq method by showing that the secretion capacity of human plasma cells is correlated with gene expression change in ER and metabolic pathways. This is a potentially interesting technical advance. However, several issues should be addressed.

1. In figure 2b, IgG secretion signals could be detected in cells positive for surface IgA and IgM, even when IgG blocking antibodies were added to the solution in the loading process, raising a specificity issue. The authors need to carefully examine whether this affects their conclusions. They should validate the secretion capacity of sorted cells using an alternative approach such as ELISPOT.
2. They need to show the IgG capturing antibodies can bind equally well to different IgG isotypes. Currently, it is unclear whether the secretion signal reflects the difference in secretion capacity or merely the difference in isotypes.

3. It is unclear whether and how frequent any cell displacement from nanovials occurs during the prolonged loading/incubation process, particularly when there are many free-floating cells judged from the FACS plots in Figure S4. In addition, cell dwelling time within nanovials will obviously influence the measurement of Ig secretion capacity. It is important to show that the loading occurs similarly across groups and cells do not exit nanovials once loaded.

4. As they pointed out, the Miltenyi's cytokine-catch assay utilizes bi-specific antibodies to link a surface protein with a secreted protein. They claim a major limitation of Miltenyi's method is the use of anti-CD45 antibody alone. They have not demonstrated that the use of anti-CD27 and anti-CD45 at the same time is better than anti-CD45 alone.

5. Other than measuring Ig secretion, can they show application of nanovials in quantitating cytokine-producing cells? This is an important issue from the perspective of technological advance, because plasma cells are known to produce antibodies in massive amounts, and it is not yet demonstrated that SEC-seq can be used to measure cells secreting factors of less abundance, such as cytokines.

Reviewer #1 (Remarks to the Author):

Summary: In this work, the authors develop a method to link secreted material from a cell (here, antibodies) with the standard 10X single cell sequencing readout. Overall the work is well done and clearly presented, and I think this technique could be useful for many applications. A few minor clarifications could be helpful.

Major Points:

R1C1: I don't feel like we have a mechanistic explanation for how NADH level alterations (or any aspect of mitochondrial biology) change antibody secretion. Can the authors speculate on this a bit? Overall (e.g. Figure 6) the authors link IgG secretion levels to several different pathways (glucose uptake, mitochondria, etc.) Some validation that the genes in these pathways functionally influence IgG secretion would be helpful.

We agree and have added a new paragraph in discussion regarding mitochondrial biology affecting plasma cell and antibody secretion. Several prior experiments in the literature are consistent with what we found regarding the links between oxidative phosphorylation and antibody secretion by PCs (please refer to PMID: 29898388).

We have also provided some additional validation studies for CD59 (see new Figure S12). We found CD59 is one of top correlates between mRNA reads and SEC-IgG barcode reads (Figure 6), as well as top in DEG (Figure 7c). We validated the presence of the CD59 protein by flow cytometry showing it marks a population of IgG secreting PCs. This new data is provided in Supplemental Figure S12. CD59 was found to be associated with insulin secretion in beta cells (PMID: 24726385) and little is known about its role in plasma cell biology. We propose that CD59 could be a good marker for PCs that highly secrete antibodies, along with other canonical PC markers, CD38 and CD138.

R1C2: I'm curious to know whether the authors could have obtained the same information by simply measuring cytoplasmic/ER or surface-bound IgG protein rather than encapsulating the secreted antibody in a hydrogel. A comparison to the enriched gene sets observed by cytoplasmic/ER expression (antibody binding) vs. secreted IgG would be interesting in any case to determine whether there are really genes that selectively influence secretion mechanism (rather than just translation)

We agree with the reviewer that, as presented, it is possible that the SEC-seq measurement is driven primarily by IgG translation. However, because IgG plasma cells do not express IgG on the cell surface, it is tricky to determine whether that is true; the transmembrane form of the antibody is removed by splicing in differentiated plasma cells. However, we attempted to partially address this comment by using two color flow cytometry to sequentially image IgG in nanovials followed by fixation of the plasma cells, permeabilization and staining for intracellular IgG. Unfortunately, the intracellular stain resulted in substantial labeling of the nanovials that is also detectable in empty nanovials, which limited our ability to interpret these results. Despite

this background labeling, some loaded nanovials exhibit higher levels of intracellular detection, especially in the secreted-IgG high group (top right quadrants, rFig 1a). We also observed cells with intracellular IgG labeling, without secreted-IgG (top left quadrants, rFig 1a- b). Although these data aren't suitable for publication, they do imply that the intracellular stains for IgG do not uniformly predict the level of protein secretion.

a.
Cell gate from DN of IgM and IgA

Intracellular IgG +
but not secreted IgG Donor43

Donor51

b.

Intracellular IgG staining
(after permeable cell)
Nanovial IgG staining
(before permeable cell)

Cell gate from DN of IgM and IgA

Reviewer Figure 1: Two color flow cytometry to image IgG secreted and bound to nanovials and intracellular IgG. (a) Flow plots of secreted IgG and intracellular IgG, gating from DN (of IgM and IgA) from two individual donors. (b) Phenotype of CD38 and CD138 from quadrant gate of (a).

Minor Points:

R1C3: Fig. 1: a bit more detail in labeling would help here. for example, what are M, A, G populations in the lower right?

The figure has been updated with labels that we hope clarify how SEC-seq will be used in these studies.

R1C4: In the single cell seq data, did the authors observe correlation between high IgG transcript level and high IgG protein level? Is there a strong correlation when the authors look at their data? Is there any previous literature on the correlation between IgG transcript level and protein level? this would help us understand the level of noise in their workflow.

We agree that the correlation between IgG mRNA transcripts and SEC-IgG barcodes is a critical question to address. In the updated figures 6 and 7, we have attempted to clarify the degree of these correlations. As expected, mRNAs for each of the IgG isotypes (IGHG1, IGHG2, IGHG3 and IGHG4) is among the most highly correlated with secreted IgG in all donors tested (Fig. 6b, 6c). The data for the individual cells are now plotted in Fig. 7a. We see significant positive correlations, but with relatively modest r values.

Reviewer #2 (Remarks to the Author)

Cheng and colleagues present an article entitled 'SEC-seq: Association of molecular signatures with antibody secretion in thousands of single human plasma cells'. Herein, the authors describe a technique that combines secretion with transcriptomic analysis. This is very interesting as a potential technique for B cell and antibody research, and the interest goes beyond – the simplicity of the workflow is promising. Therefore, in terms of technology, the approach is novel and promising and might be of interest to the readership of Nature Communication. The authors study ex vivo stimulated B cells from healthy donors and correlate a proxy of secretion with the transcriptome of the cells. These experiments recapitulate findings from the B cell field but also show some interesting new findings. Here, however, the experiments performed and presented seem preliminary and, while showing interesting initial findings, are far from displaying conclusive biological data and the vigor of data I would expect from a publication in Nature Communications.

R2C1: Firstly, many experiments seem to be done with one donor (Figure 1d) or two donors (Figure 1c) only. This statement is also extracted from the conclusion (as the authors state that

3000 cells can be analyzed in a run, and Figure 4c shows 3060 cells) that the experiment has been performed once on one donor. This drastically limits the biological meaning of the results.

We thank the reviewer for this comment and have repeated the study with cells from 2 additional donors for Figure 2d/2c, and have included these along with the results in the initial submission. The legend now reads Figure 2c n=8 (4 unique donors)/Figure 2d n=7 (3 unique donors). For the SEC-seq studies, we now have included data from ~13000 cells from 3 different donors. All in all, the addition of this new data greatly strengthens our conclusions and has enabled us to demonstrate the robustness of the assay (see Fig.'s 6b, 7b, and 7c).

R2C2: Second, the introduction states, 'No current technology has been able to link the amount and type of secreted molecules from a single cell with its transcriptional profile, nor has this type of association been multiplexed at a scale of hundreds to thousands of single cells simultaneously.' This statement is not exactly valid as it neglects a variety of work from different research groups and companies – the authors present the system from Berkeley Lights and Isoplexis, but many publications have been published over the last years linking the level of secretion to the sequences (which is a similar level of quantitation as the study provides). Therefore, I wonder if the presented technology really overcomes these trade-offs. While still innovative and interesting, this, in my view, decreases the added value of the method. A combination of flow cytometry and nanovials has already been published.

We are unsure of which prior studies the reviewer is referring to. We did find the following citation (PMID 27626628). In the work of Wang et al., ~20 individual cells were analyzed using a custom microwell device with a top plate. Secretions from individual mouse macrophages were captured on a top plate where they could be stained and associated with transcripts by manually picking cells from the microwells and placing them into a 96 well plate for single-cell RNA sequencing. Using the SEC-seq approach we demonstrated a two orders of magnitude increase in the number of cells that can be analyzed. We are not convinced that the prior studies allow for the scale or multiplexing to the same degree as SEC-seq. In addition, SEC-seq is easily accessible using equipment already in labs, which should enable more researchers to use the technique.

R2C3: Thirdly, the authors should clarify in their manuscript whether they study 'human plasma cells', primary or ex vivo stimulated. As it stands, it is often left open to interpretation, and what we can learn from each system is fundamentally different.

All the work in this manuscript was done using primary *ex vivo* differentiated plasma cells. We have clarified the source of cells in line 305-306. As we have previously demonstrated and cited herein, *ex vivo* derived plasma cells exhibit many features observed in human plasma cells isolated directly from bone marrow including transcriptional profile, scanning electron micrograph ultrastructural features, protein production capacity and the ability to home to and be retained in bone marrow. Please peruse Cheng et al, Nature Communications 13, 6110 (2022) for more information (doi.org/10.1038/s41467-022-33787-8). Based on our *in vivo* data, this *ex*

vivo system produces cells that faithfully recapitulate many aspects of human plasma cell biology.

R2C4: Additionally, no comments about efficiencies are performed – if I start with 1000 B cells secreting antibodies, what can I expect at the end? If I do the math with the information provided, only a small fraction will be analyzed successfully. This would be a major limitation (or at least a point) worth mentioning and discussing. Over the figures, it is very difficult to understand how efficient the whole process is, from a cell culture of a certain number of cells to the transcriptomic end. In general, many figures describe the process (or show examples/illustrations) but give little quantitative information – this is necessary if you want to convince readers to use this technique. Many relevant controls are missing – what is the frequency of ASCs in your *ex vivo* stimulated cells? Does this agree with your measurements? How were these findings controlled?

We thank the reviewer for this suite of comments and have added text to show potential efficiency of this process. In summary, we estimate that ~30% of the input population can be successfully loaded into nanovials. And ~70% are viable (after Ficoll). And for GEM loading, we prepare 2x extra samples (~50% in reaction). After handling, which includes multiple washing steps, spinning, straining, filtering, sorting, and passing cellranger QC, we lose an additional 60-70 % of cells from all those processes. So, we estimate that one would need to start with ~100K cells in order to analyze 4,000 cells using 10X genomics. We added this information to the updated Figure 4a.

Regarding the frequency of ASCs in the *ex vivo* cultures, please look at the gating in Figure 2B-C. As can clearly be seen by the reader, using the culture methods described here and in Cheng et al, Nature Communications 13, 6110 (2022) (doi.org/10.1038/s41467-022-33787-8), and with our phenotypic data, we observe that ~60% of the cultures exhibit a CD38^{hi}CD19^{lo} phenotype that we characterize as plasmablasts. From that population ~30% of the cells express CD138, a marker of plasma cells. Phenotypic historical data is shown below (Reviewer Figure 2).

Reviewer Figure 2: Phenotype of ex vivo differentiated ASCs. Left bar graph was the percentage of cells gated from live cells, the right bar graph was the percentage of cells gated from ASC cells (CD38+), n=6.

In our prior study, we showed that greater than 50% of the bulk population exhibit ER structural features and mRNA signatures consistent with them being plasma cells, which we have also correlated with ELISPOT results.

Finally, it is not clear what additional “controls” the reviewer would like. Within this heterogeneous population that we have extensively characterized in prior work, there are cells that produce IgG and those that don’t (e.g. activated B cells or IgM/IgA plasma cells). As expected, these “control” populations exhibit less detectable secreted IgG as measured by either flow cytometry, AMNIS or SEC-seq.

R2C5: I also challenge the statement that this technique quantitates secretion – I am pretty convinced that you can see differences between secreting and non-secreting, but the text often refers to these two subclasses only and also says from time to time 'level of secretion'. A level and quantitation are very different, and many more controls are needed to make such a statement (as various techniques allow quantitative secretion analysis down to the molecules/s). Throughout the manuscript, the authors call cells low and high secreting – it seems not clear why you have two groups of cells, and looking at figure 5c, it looks more like a continuous log Normal distribution and not two distinct populations. Also, the legend calls the cells (non-secreting), whereas the figure and text refer to the same cells as low IgG – if they are low IgG, this would also mean that most IgA and IgM are low IgG secreting? Here, I would rather suggest that some IgG-secreting cells and cells do not secrete. Also, there is also a considerable amount of IgM and IgA cells above the threshold of log5 – what does this mean?

We thank the reviewer for this comment and agree that this method does not assess the absolute quantities of IgG being produced by each cell. The best use of this approach is to compare the relative protein secretion of cells within a heterogeneous population. We have made edits to the text to emphasize that this system is being used to measure relative levels, not absolute quantification.

Defining secretion thresholds (low/high vs/ secreting/non-secreting cells) requires a little subtlety due to the way we can assess doublets (ie 2 cells in one nanovial), and the degree of background observed in the method. To address the meat of this concern, we have eliminated Fig. 5c and took the reviewers' suggestion of using a more simplified approach to describe secretors. For nomenclature, we simply refer to the IgG barcode (SEC-IgG high and SEC-IgG low), rather than making a determination about whether the cells with low barcode detection are low-secretors or non-secretors. See our comments below about doublets and background:

Doublets: While doublets are clearly visible/addressable in Amnis, in the 10X genomics studies, doublets can only be assessed informatically based on a UMI threshold, or by co-expression of multiple antibody isotypes. However, because cells can fall out during the handling of the nanovials (see figure in response to **R3C3**), we suspect some nanovials initially containing doublets will only retain one cell in the 10X analysis and not be eliminated informatically. This possibility could result in nanovials with IgA/M+ cells that exhibit high SEC-IgG barcodes. As expected, these are quite rare in the full analysis. See also the response to **R2C11** for a more detailed explanation about the analysis..

Background: We agree with the reviewer that a subset of IgG cells DO NOT secrete antibodies - see the response to reviewer 1 (**R1C1**). However, using the 10X method, we cannot definitively determine which cells are non-secretors versus those that secrete low amounts. This is because the barcode oligos and/or the conjugated antibody nonspecifically interacts with the cell or nanovial substrate (PMID: 33589839). This leads to a relatively substantial signal in cells that do not secrete IgG (eg IgA/IgM plasma cells). This is a limitation of the SEC-seq approach, and CITE-seq in general.

In summary, while I appreciate the simple approach and the value of the technique, considerably more experiments, and controls are needed to conclude the biological data. As the experiments will take some time, I would reject the current version of the manuscript, but I remain available if the data is added.

A few minor comments from my side:

R2C6: I think it needs to be clear that these are ex vivo stimulated human B cells – this makes a huge difference as we would expect a single Gaussian distribution in terms of secretion (and one

We thank the reviewer for this comment and have adjusted the text to clarify that these cells are primary human B cells that we have differentiated into ASCs. As described in the response to **RC2C4**, the majority of the analyzed cells are no longer B cells (naive B or activated B).

R2C7: Why are viabilities so low? In our experience, you should expect 80-90% viability after 10 days of ex vivo stimulation, yet the data shown is somewhere between 14-50 % for loaded vials – is this viability sufficient for sequencing? I understand that only viable cells are sorted, but such an environment will certainly have an influence on secretome and transcriptome.

We agree with the reviewer that *ex vivo* stimulation of human B cells with cytokines that promote an activated phenotype (CD40L, CpG, IL21), will retain high viability following 7 days in culture. However, following removal of these cytokines and addition of cytokines that promote differentiation (see methods section and our prior papers referenced above), many of these cells die. This is a well-established phenomena that happens during plasma cell differentiation - for example, see the following review: doi: [10.3389/fimmu.2018.02053](https://doi.org/10.3389/fimmu.2018.02053) and primary literature: DOI: [10.1182/blood-2009-07-235960](https://doi.org/10.1182/blood-2009-07-235960)

However, to clarify the flow chart shown in Fig. 4a, the nanovials were loaded immediately after a Ficoll purification and start with a ~70% viability, which we propose is a suitable environment for the assay. The post-assay analyses we show in Figure S1B are from loaded nanovials. Note that the dyes used in these experiments exhibit some binding to the nanovials. We found that analysis of the top-left quadrant of these distributions yielded higher quality labeling with the cell surface markers, which is consistent with this subset being viable. We used this conservative gating for downstream analyses (see the numbers listed in response to **R2C7**), but are confident that this gating greatly underestimates the overall cell viability. Finally, in the sequencing study, we sorted live cells from loaded nanovials. Therefore, the viability was >90% in the 10X reactions and allowed us to generate high quality 10X data as can be seen in Fig. S8.

R2C8: Figure 1c: Why one-way instead of two-way? The figure also shows 6 points, the legend 5.

Since there is a single variable being compared over multiple conditions, one-way analysis of variance is an appropriate test for the comparison referred to by the reviewer. While this study has relatively small numbers, the distributions appear normal and similar across groups, which are the primary considerations when determining whether one-way ANOVA is appropriate. We are not sure how a two way ANOVA could be used in this context.

We found that 2 points were stacked in the visualization of the CD38+ data; this figure has been updated with 2 extra donors (now 8 data points) - in the other 2 measurements, 6 points are clearly visible in the original panel.

R2C9: 'Upon looking at the percentage of IgG secretors in each subcluster, we found that IgG clusters have on average ~ 25% IgG secretors and ~ 40% non-secretors, whereas the remaining clusters were predominantly non-secretors.' How does this correlate with the statement from before that between 20-70% of cells were ASCs in the cytometry experiments?

We thank the reviewer for the comment. We would like to point out that 25% (with the new data taken into account, the number is 31-32%) is within the range of 20-70%. However, as

described above in the response to **R2C5**, the flow cytometry and SEC-seq assays are not directly comparable, due to the caveats associated with detection of the IgG-barcode (background, doublets).

R2C10: 'Approximately 12% of the loaded nanovials were IgG positive in the cell sort (Fig. S8a, right panel), which is comparable to the ~14% of IgG secretors we quantified in the scRNA sequencing data.' – 12% are positive in the sort, but 25 % are IgG secretors in the subclusters? How does this work?

We thank the reviewer for pointing out our lack of clarity. Our point that 25% of IgG clusters exhibit barcode counts that we defined as “IgG secretor” (now as SEC-IgG hi) is describing the proportion of cells that express an IgG transcript that also were defined as an “IgG secretor”. It's important to note that ~35% of the cells were NOT in IgG subclusters (i.e. these cells expressed RNA for IgA, IgM or no antibody isotype). When you take this into account, we hope the reviewer can appreciate how we concluded that 14% of the cells analyzed by RNA sequencing (transcripts for IgG, IgA, IgM, no isotype) were defined as an “IgG secretor” using our barcode distribution metric and that 25% of IgG positive cells analyzed by RNA sequencing (only transcripts for IgG isotypes) were defined as an “IgG secretor”(SEC-IgG hi). Thus, the flow results using a fluorescent readout are consistent with the conclusions we reached using the barcode readout in the sequencing study. We have clarified this in the updated manuscript.

R2C11: As the Raji/Hybridoma experiments show multiple loading (cross-contamination) of up to 9%, what does this mean for the sequencing experiments of B cells? Should I expect a similar background of multiplerts?

We thank the reviewer for this question. The short answer is that we do expect similar numbers of doublets in sequencing study. We have taken 2 approaches to ELIMINATE potential doubles from the SEC-seq analysis. First, we used standard scRNA sequencing analysis methods to eliminate doublets based on being outliers in the UMI distribution (i.e. we eliminated cells with too many UMIs). Second, since we know that individual antibody isotypes and light chains are not typically co-expressed in B cells, we had expression thresholds for each of these molecules and eliminated cells that co-expressed either >1 unique light chain (Kappa, lambda 1-3), or >1 heavy chain isotype (IgA1, IgA2, IgM, IgG1, IgG2, IgG3, IgG4 or IgE). In the published analysis, there should be very few doublets (<10 total cells).

R2C12: Acknowledges: 'other members' – either names or remove

This has been updated with specific names.

R2C13: If I calculate the numbers in SI Figure 1b, I get much lower frequencies – please double check. Also, the viability is very low – do the nanovials fish out mostly dead cells or is the general viability very low in the culture? Both have implications for your results... this is not discussed.

The images in Figure S1B are of loaded nanovials. Note that this is a critical distinction because the dyes used in these experiments exhibit some binding to the nanovials. Therefore the dye used here does not clearly discriminate between live-dead within the loaded nanovial populations. We found that analysis of the top-left quadrant of these distributions yielded higher quality labeling with the cell surface markers, which is consistent with this subset being viable. Thus, for our analyses of the surface markers, we used these conservative viability gates despite the fact that they likely underestimate the viability of the cells in nanovials. Please see the response to **R2C11** for a discussion of cell viability in plasma cell cultures.

R2C14: As a remark to Figure S2: Larger spots could also be to larger cells? I know that this is often done in ELISPOT, but I am always wondering about the potential bias in these analyses – why are two classes displayed? How are they defined? Should an integral of a larger spot not anyway be larger if not corrected by the size?

We thank the reviewer for this interesting/insightful point. Figure S2 is a representative figure included to illustrate that there is variance in spot size by ELISPOT. Of course this variance in relative ELISPOT detection could be driven by isotype affinity, cell size, metabolic potential, secretory fitness or all of the above. Note that we also observe variance of 10X IgG barcode detection within IgG cells expressing the same antibody isotype, which indicates that isotype affinity is not the sole reason for differences in the detection of IgG secretion; different cells are highly likely to produce different quantities of antibody.

In Figure S2, we are showing representative spots (green and red) to make the very point you are making that ELISPOT assays do not provide sufficient information to discern any features of the secreting cells; whether it's size, or other characteristics. In addition to the representative spots, we show the distribution of the pixel intensity of all the spots on the right; this is the relevant information. It isn't clear what additional definitions are required for this representative and extremely qualitative example. We will change the text to make this point more explicit.

R2C15: Figure S3: without axes and nothing, what do I see?

Thanks for the commenting, we've updated the axis to make it clear that this is a normalized count of IgG secretion on nanovial fluorescence signals (y-axis).

R2C16: Nature Portfolio Report: How can you calculate sample sizes of 1 and 2, or 3'000? How was this done?

Thank you for making this point. In the updated Figures and legends, we clarified how the sample sizes were calculated and only indicate the number of independent experiments and donors. In the initial submission, we included the number of cells analyzed, which is likely the source of this confusion.

R2C16: Gating strategy: How was the gate for living/dead chosen? Why does CD19 expression in the gate vary so highly?

Please refer to the responses to **R2C11** and **R2C13**. Here, we show a zoomed-in panel of live dead contour plot (x: live-dead, y: SSC-A) to demonstrate how we gate the live cells based on the contour.

Reviewer Figure 3: Contour plot of lived dead (Indo-violet) vs SSC-A.

We agree that the CD19+ plots from our earliest studies were confusing. Because those were not critical to the interpretation of the results, the data has been removed.

R2C17: Lastly, the flow cytometry, in contrary to the reporting chapter, are not displaying axis, numbers and fluorophore (for example, figure 1b, 1c, 1d).

We thank the reviewer for this comment and assume it is in reference to Figure 2. We have added numerical labels to indicate the numbers that show up on the flow plots in b, c, and d, most importantly the log ticks. And fluorophores are updated in each axis. We have additionally added nomenclature to Figure 2 for clarification.

R2C18: Within the discussion, data are discussed that are not shown – please remove this statement or show the data. Lastly, the discussion also does not discuss the potential limitations of the technique.

We thank the reviewer for this comment. The mesenchymal stromal cell data discussed regarding VEGF secretion has now been cited. Additionally, we have added a section on limitations to the discussion. For reference, <https://doi.org/10.1101/2023.01.07.523110>

Reviewer #3 (Remarks to the Author)

In their previous studies, James and colleagues developed bowl-shaped hydrogel particles, termed “nanovials”, that allow capturing of secreted proteins from single cells and are compatible with flow cytometry analyses. In the current manuscript, the authors extended previous work to show that the nanovial system is compatible with the commercially available single-cell mRNA sequencing workflow. They further provided proof-of-principle demonstration of the SEC-seq method by showing that the secretion capacity of human plasma cells is

correlated with gene expression change in ER and metabolic pathways. This is a potentially interesting technical advance. However, several issues should be addressed.

R3C1: In figure 2b, IgG secretion signals could be detected in cells positive for surface IgA and IgM, even when IgG blocking antibodies were added to the solution in the loading process, raising a specificity issue. The authors need to carefully examine whether this affects their conclusions. They should validate the secretion capacity of sorted cells using an alternative approach such as ELISPOT.

We thank the reviewer for this comment. Regarding the expression of IgG within the IgM/A populations, the original figure 2b was not done using blocking reagents. We highlighted the effect of blocking reagents during the loading process in Figure S3. Also, in prior studies we have sorted plasma cells using surface IgM and performed ELISPOT. With this method, we did not see IgG spots. This data can be provided upon request.

Therefore, we suspect the residual IgG false positive in IgM/A is primarily due to cell doublets (see response to **R2C5**). After modifying the gating strategy FSC-A and FSC-H, we found the IgG secreting IgA/M+ cells to be ~6-16%, which is similar to the doublet rate that we calculated experimentally in Figure 3 (~10%). Representative data using the blocking reagent is presented below and has been added to the figure 2b.

Figure 2b: Representative flow cytometry density scatter plots for surface markers to identify populations of ASCs from active B cells using CD19 and CD38 staining. IgA cells, IgM cells, and ASCs not producing either IgA or IgM (double negative, DN) were gated based on IgA and IgM staining. Fluorescence histograms of IgG secretion signal for the various identified gates and empty nanovials containing no cells

Finally, for the flow experiments, we don't think that IgA/G, or IgM/G doublets impact the analysis because we are only analyzing the double-negative population. However, it's feasible

that doublets containing cells expressing multiple IgG isotypes could be present at some frequency (~10%). These could contribute to the differences in relative secretion between cells, but we have no reason to believe that the doublets would be enriched in any of the specific phenotypes discussed in 2c-d. Additionally, the Amnis data explicitly eliminates doublets visually and led to similar conclusions as the data using flow.

R3C2: They need to show the IgG capturing antibodies can bind equally well to different IgG isotypes. Currently, it is unclear whether the secretion signal reflects the difference in secretion capacity or merely the difference in isotypes.

We thank the reviewer for this question. Differences in isotype-specific affinity of our reagents could be one potential reason for differences in the SEC-IgG hi populations observed across cells expressing different isotype transcripts (especially IGHG3). This potential factor along with differences in secretion levels that are isotype-specific could result in different amounts of SEC-IgG signals. Since no further investigations were based on this difference, we now clearly lay out the limitations in interpreting this minor result of our study (line 533-549).

R3C3: It is unclear whether and how frequent any cell displacement from nanovials occurs during the prolonged loading/incubation process, particularly when there are many free-floating cells judged from the FACS plots in Figure S4. In addition, cell dwelling time within nanovials will obviously influence the measurement of Ig secretion capacity. It is important to show that the loading occurs similarly across groups and cells do not exit nanovials once loaded.

Because we incorporate capture antibodies on the surface of the nanovials, and nanovials have a protective cavity which shields cells from fluid shear stresses once cells are loaded we see very few cells are dislodged during the process of cell incubation and analysis. The majority of background free cells are instead cells that were never loaded onto the nanovials or filtered out during our procedures. If large numbers of secreting cells were being dislodged, we would observe this in our imaging flow cytometry data sets as empty nanovials with high levels of IgG signal. We re-visited our data and found only 10% of IgG positive nanovials had no associated cell, suggesting low cell displacement during processing steps. Figure for reviewer is shown below. We also want to emphasize that dislodged nanovials first gated out in flow analysis, only cells with surface markers are in analysis (CD19+). And SEC-seq dislodged cells aren't considered as cells by Cellranger, Cellranger QC ruled out the datapoint before we analyze it.

Reviewer Figure 4: Amnis analysis of empty nanovial with IgG signal.

Upper panel shows the event gated from an empty nanovial with IgG signal. And the bottom panel shows the gating strategy and gate percentage (10% of dislodged nanovials).

R3C4: As they pointed out, the Miltenyi's cytokine-catch assay utilizes bi-specific antibodies to link a surface protein with a secreted protein. They claim a major limitation of Miltenyi's method is the use of anti-CD45 antibody alone. They have not demonstrated that the use of anti-CD27 and anti-CD45 at the same time is better than anti-CD45 alone.

Please note that the nanovial assay is not limited to specific cell types. anti-CD27/CD45 were used to capture plasma cells in this work, but different antibodies, or even extracellular matrix proteins, can be used to capture other cell types. This cell capture approach on nanovials is also independent of the type of secretions captured. The key limitation with the Miltenyi cytokine-catch assay is that both the cell type specificity and secretion capture antibody are linked together in one reagent, a bispecific antibody, limiting the flexibility of the platform. For example, there are no reagents that are available to bind to ANY surface marker and capture secreted IgG, which would be necessary for this work. We make the limitations of the catch assay more clear in the revised introduction (line 91-92).

R3C5: Other than measuring Ig secretion, can they show application of nanovials in quantitating cytokine-producing cells? This is an important issue from the perspective of technological advance, because plasma cells are known to produce antibodies in massive amounts, and it is not yet demonstrated that SEC-seq can be used to measure cells secreting factors of less abundance, such as cytokines.

Although additional studies measuring cytokines are outside of the scope of the current work, we expect SEC-seq should also be applicable to cytokine-secreting cells. Please see our now accepted preprint (PMID: 36711524) showing the ability to detect cytokines, such as IFN-gamma, IL-2, and TNF-alpha, from human T cells using the nanovial platform. Although the data

shows results using fluorescently-labeled antibodies, these antibodies could be oligonucleotide labeled as is required for the SEC-seq workflow; this work is now cited in line 523 and 540 (reference 28 and 34) .

REVIEWERS' COMMENTS

Reviewer #1 (Remarks to the Author):

the authors have addressed most of the suggestions, it's a nice method, and I think the paper should be published.

just to clarify, regarding my point #2 I was wondering if you were to not use nanovials at all, and instead just flow sorted after staining fixed cells for IgG production (in the ER, in cytoplasm, on surface), would you see the same RNA expression patterns by doing standard single cell sequencing? This is trying to get at what we are learning specifically by being able to encapsulate a cell and monitor secretion.

The authors tried to do intracellular staining in nanovial-encapsulated cells and this didn't work too well, which is fine. The authors suggest there are cells that are IgG positive inside that don't show nanovial staining, but the correlation looks fairly high overall. In any case, I don't think this is a problem for the story necessarily, but for future applications one might want to be able to distinguish protein production from true secretion, so the authors might add some notes about how to address this.

Reviewer #2 (Remarks to the Author):

I thank the authors for submitting their reviewed manuscript. They have added an impressive amount of work and data following the first revision - indeed, all of my initial concerns and requests have been answered, and I can see that some of my comments were also unfounded due to initial misunderstandings from my side, points that have been clarified. The manuscript has gained in clarity, I especially appreciate the new figures with added numbers that make following the approach, its usefulness and expected outcomes much clearer. I appreciate the thorough and clear revision, and while reading the manuscript, everything seems clearer to me now - an issue I had during my first response.

Therefore, I would suggest to accept the manuscript as it is.

A small note from my side:

Following your answer on comment 2, I went back to the respective literature in detail. I have to agree with the authors - while many individual aspects are present in literature (for example, the high-throughput analysis of VH/VL sequences upon secretion), no current study fulfills all the criteria outlined for the novelty... the combination of transcriptomal profile, throughput in combination with secretion.

Therefore, I would like to take back this comment and excuse my initial statements that was based on a misconceptualization on my side.

Reviewer #3 (Remarks to the Author):

They have satisfactorily addressed my concerns.

Reviewer #1 (Remarks to the Author):

The authors have addressed most of the suggestions, it's a nice method, and I think the paper should be published.

Just to clarify, regarding my point #2 I was wondering if you were to not use nanovials at all, and instead just flow sorted after staining fixed cells for IgG production (in the ER, in cytoplasm, on surface), would you see the same RNA expression patterns by doing standard single cell sequencing? This is trying to get at what we are learning specifically by being able to encapsulate a cell and monitor secretion.

The authors tried to do intracellular staining in nanovial-encapsulated cells and this didn't work too well, which is fine. The authors suggest there are cells that are IgG positive inside that don't show nanovial staining, but the correlation looks fairly high overall. In any case, I don't think this is a problem for the story necessarily, but for future applications one might want to be able to distinguish protein production from true secretion, so the authors might add some notes about how to address this.

We thank the reviewer for this comment/clarification, and for prior suggestions that greatly improved the study. Please note that after fixation and permeabilization required for intracellular staining, there is a significant loss of mRNA, which prevents high quality single-cell sequencing. Furthermore, even if we could obtain data, we predict that IgG might not be the best system to highlight this difference between cells that have translated, but not secreted protein. For example, a better system may be studying eosinophils which are poised for secretion of the RANTES chemokine (i.e. retain RANTES intracellularly that isn't secreted; PMID 10381494), and some are actively secreting. In this example, if we were to just look at intracellular staining, we couldn't discriminate between the two possibilities. This situation would be a definitive use case in support of testing this idea using nanovials. We will prioritize this idea as we continue our work.

Reviewer #2 (Remarks to the Author):

I thank the authors for submitting their reviewed manuscript. They have added an impressive amount of work and data following the first revision - indeed, all of my initial concerns and requests have been answered, and I can see that some of my comments were also unfounded due to initial misunderstandings from my side, points that have been clarified. The manuscript has gained in clarity, I especially appreciate the new figures with added numbers that make following the approach, its usefulness and expected outcomes much clearer. I appreciate the thorough and clear revision, and while reading the manuscript, everything seems clearer to me now - an issue I had during my first response. Therefore, I would suggest to accept the manuscript as it is.

A small note from my side:

Following your answer on comment 2, I went back to the respective literature in detail. I have to agree with the authors - while many individual aspects are present in literature (for example, the high-throughput analysis of VH/VL sequences upon secretion), no current study fulfills all the criteria outlined for the novelty... the combination of transcriptional profile, throughput in combination with secretion. Therefore, I would like to take back this comment and excuse my initial statements that was based on a mis-conceptualization on my side.

Thank you – we appreciated the effort you have taken to provide comments to improve the manuscript.

Adding replicate data to the scRNA sequencing studies has greatly improved the robustness of the analysis.

Reviewer #3 (Remarks to the Author):

They have satisfactorily addressed my concerns.

Thank you – we appreciated the effort you have taken to provide comments that have greatly improved the study.